# Level Up: Defining and Exploiting Transitional Problems for Curriculum Learning

## Abstract

Curriculum learning, ordering training examples in a sequence based on difficulty, takes inspiration from human learning but has not gained widespread acceptance. Static strategies for scoring item difficulty produce curricula that are not specific to the learner at hand, and that rely on indirect proxy scores of varying quality. Dynamic approaches base difficulty estimates on gradient information, requiring considerable extra computation during training. We introduce a novel method for measuring the difficulty of individual problem instances directly relative to the ability of a given model, and identify *transitional problems* that are consistently easier as model ability increases. Applying this method to chess and mathematics, we find that training on appropriately calibrated problems most efficiently "levels up" a model to the next competence tier. These problems induce a natural progression from easier to harder items, which outperforms other training strategies. By measuring difficulty directly relative to model competence, our method yields interpretable transition problems, learner-specific curricula, and a principled basis for step-by-step improvement.

## 1 Introduction

Machine learning (ML) differs significantly from human learning in practice, both in terms of the learner and the learning method. Human learning is generally curriculum-driven, with a training diet of examples that increase in complexity as the learner improves in ability. On the other hand, an ML model optimized on data drawn i.i.d. from the training distribution (i.e. empirical risk minimization, or ERM) typically learns to perform a task at least as well as when optimized on the same examples presented in a structured manner. Curriculum learning (Elman, 1993; Sanger, 2002; Bengio et al., 2009)—the process of training ML models on data that are progressively more difficult to learn—has shown benefits in some settings, particularly for models that are trained on noisy or limited data (Wu et al., 2020; Wang et al., 2023), in models for sequential decision making such as in reinforcement learning (Tao et al., 2024; Zhao et al., 2022), and in the post-training of foundation models for mathematical reasoning (Zeng et al., 2025). However, several negative results for curriculum learning temper these successes. Curricula have been observed to show no benefit in end performance over ERM on high-quality or very large multi-task datasets (Wu et al., 2020), with overparameterized networks (Mannelli et al., 2024), and even when training on data that emulates human learning (Warstadt et al., 2023).

Though learning from data sampled i.i.d. from the training distribution remains the de facto standard recommendation in ML, we observe that training with various forms of non-uniform data ordering is standard in modern ML pipelines. Given the considerable computational expense of large-scale training, methods that improve sample efficiency through strategic data selection and ordering become valuable considerations. GPT-3 and T5 had non-uniform mixing strategies (Brown et al., 2020; Raffel et al., 2020), and the training of more current foundation models is increasingly divided into pre-, mid-, and post-training phases (Wang et al., 2025; Ouyang et al., 2022), with different training tasks, datasets, and learning objectives in each phase (Liu et al., 2024; Zeng et al., 2025).

Given the mixed bag of results, the prevalence of curricula in frontier model development drives us to reassess the structure and implementation of curriculum learning for training ML models. A primary consideration for curriculum learning involves defining an appropriate measure of difficulty to order training examples from easy to hard. Methods of measuring the difficulty of training examples are

either designed for a specific domain (e.g., in BabyLM by Oba et al. (2023)) or dependent on the training dynamics of a model (e.g., Graves et al. (2017)). Domain-specific curricula often rely on human-centric measures that do not translate well to the difficulty of learning examples with ML models, while measures that depend on training dynamics (such as the reduction in loss or gradient norm) require considerable computation during the training process.

In this work, we are inspired by the common practice of curricula for human learning to focus on getting learners to 'level up' (e.g., a $4^{\text{th}}$ grade scholastic curriculum focuses on getting a student to a $5^{\text{th}}$-grade level). This local notion of improvement applies to a range of learners, and leverages experience as to what kinds of problems typically separate levels of competence. We ask the following questions: (1) During the process of learning a subject, are there consistencies in the kinds of problems a learner at a particular level of competence can and cannot solve? (2) Can we identify these specific problems at each level, or better yet, the characteristics of such problems? (3) Does training on these or related problems provide an efficient way for the learner to make progress towards the next level, and inform curriculum learning strategy?

In order to address the question of whether problems can be identified that characterize the level of a learner, we define *transitional problems* as problems that exhibit a sharp transition in solvability across increasing competence levels: solved by models at or above a given level, but not by those below. These problems mark competence boundaries and yield an empirically grounded easy-to-hard partitioning of the training data distribution. We study two domains—chess and mathematics— to identify transitional problems, and examine whether training on these transitional problems at the next competence level more efficiently 'levels up' a model than training on problems at other levels. We then investigate whether a curriculum with ascending difficulty over transitional problems enables step-by-step improvement in ML models, comparing its performance to other training strategies, such as curricula with descending difficulty and i.i.d. baselines. The experimental results suggest that transitional problems may provide a promising avenue in the pursuit of sample-efficient training strategies for reasoning models.

Transitional problems can be identified using existing model sequences, such as skill-adaptive families (e.g., Maia), training checkpoints, or generations of frontier models, rather than requiring a specially trained oracle. We show that training the strongest model specifically on problems that all earlier models fail yields the largest improvement, indicating that transitional problems provide a directional and sample-efficient training signal. Even when the model sequence must be trained, this up-front cost can be amortized across downstream models, domains, or tasks, as in meta-learning or personalization, and we demonstrate the transfer of transitional problems from one mathematics dataset to another via embedding similarity. Finally, transitional problems are intrinsically informative; they reveal the specific problems that mark competence boundaries, which often diverge from human-designed skill hierarchies, and can be helpful to future learners.

## 2 TRANSITIONAL PROBLEMS

Given a series of models that differ in performance on a learning task (e.g., periodic checkpoints of a model being trained on that task), we order and partition models into competence levels, where the performance of a model at level $i$ is empirically stronger than one at level $i - 1$, and weaker than one at level $i + 1$. The learning difficulty of an example is then the level of the weakest model that is correct on that example. Additionally, we enforce a *monotonicity* constraint on our training examples: any model stronger than the weakest model that answers a problem correctly must also answer that problem correctly. For a classification task, this equates to the probability of the correct class of an example at level $j$ increasing as the level of a model increases, with the correct class becoming the most probable class for all models at level $j$ and beyond. We designate examples that satisfy this constraint as *transitional problems* (Figure 1) to indicate the smooth transition in easiness as models grow in competence.

### 2.1 DEFINING TRANSITIONAL PROBLEMS

In concrete terms, we define a model series as a sequence of models of increasing strength.

**Definition 1** (Model Series). *Let $\mathcal{M} = \{M_0, M_1, \ldots, M_n\}$ be a finite set of $n + 1$ models. We say $\mathcal{M}$ forms a* model series *if there exists a strength function $s : \mathcal{M} \to \mathbb{R}^+$ such that:*

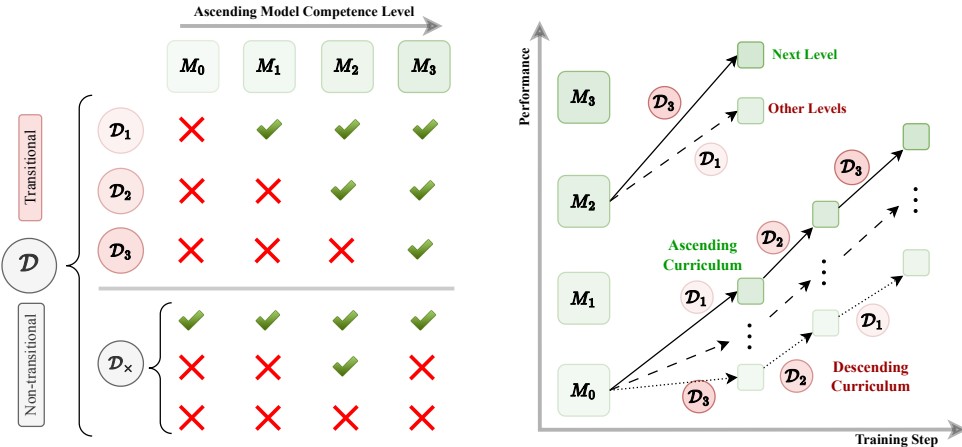

Figure 1: Transitional Problems at a level $i$ can only and consistently be solved by models at a competence level $j \geq i$. We find that training on the next-level transitional problems most efficiently "levels up" a model to the next competence level, which induces a natural ascending curriculum that can be compared to other training strategies.

- *Each model has a unique strength:* $\forall i, j \in \{1, \ldots, n\}, i \neq j \implies s(M_i) \neq s(M_j)$; *and*

- *The models are indexed by increasing strength:* $s(M_i) < s(M_j)$ *for all* $0 \leq i < j \leq n$

*We denote the strength of model $M_i$ as $s_i = s(M_i)$. The set $\mathbb{R}^+$ represents the positive real numbers.*

Given a model series $\mathcal{M}$ with a strength function $s$, we define the *transitional problems* of $\mathcal{M}$ with respect to a data distribution $\mathcal{D}$ as follows.

**Definition 2** (Transitional Problem). *Let $\phi_p(M_i) = 1$ if model $M_i$ correctly solves problem $p$, and $\phi_p(M_i) = 0$ otherwise. A problem $p$ is called a* transitional problem *if there exists a unique $k \in \{1, \ldots, n\}$ such that:*

- $(\forall i \in \{0, 1, \ldots, k-1\} : \phi_p(M_i) = 0) \wedge (\forall i \in \{k, k+1, \ldots, n\} : \phi_p(M_i) = 1)$

*Note that $k \geq 1$ ensures at least one model ($M_0$) fails before the transition. We call $\tau_p = k$ the* transition point*, $M_k$ the* transition model*, and $s(M_k)$ the* transition strength *for problem $p$.*

**Definition 3** (Transitional Problems at $\tau$). *Given a data distribution $\mathcal{D}$ and a transition point $\tau \in \{1, \ldots, n\}$, the transitional data distribution from $\mathcal{D}$ w.r.t transition point $\tau$ is defined as:*

$$\mathcal{D}_\tau = \mathcal{D} \mid [(\forall i < \tau, \phi_p(M_i) = 0) \wedge (\forall i \geq \tau, \phi_p(M_i) = 1)]_{p \sim \mathcal{D}} \tag{1}$$

Restricting the training distribution to transitional problems produces a *partially ordered* training set where problems at a given level are equivalent, but are strictly harder (easier) than the problems at a lower (higher) level. Notably, this difficulty is with respect to the model series and not based on a human-centric metric. A curriculum that starts from the next level of a learning model is thus a true representation of an easy-to-hard ordering for that model, and avoids training on problems that are trivial or intractable.

## 2.2 TRANSITIONAL PROBLEMS IN CHESS

**Chess as a Model Domain.** Our study requires a series of models of monotonically increasing strength that coherently captures the progression of competence, allowing us to identify transition points where specific problems shift from unsolvable to solvable. Game-playing, especially chess, is an ideal model domain for this setting for the following reasons. Chess has a massive and diverse human player base with a standard metric for evaluating competence—the Elo rating (Elo,

1978). Chess is also well-studied in the AI literature, with a range of strong models and a massive database of human and computer games with extensive metadata. Thus, we focus our analysis on the acquisition of chess proficiency, with additional results in the mathematical reasoning domain.

**Choosing a Chess Model.** Our analysis requires a model series that can play chess at multiple levels of competence and be trained to improve. Although traditional chess engines such as Stockfish (Romstad et al., 2023) and AlphaZero (Silver et al., 2017) can vary their search depth to weaken play from superhuman levels, this adaptation is not well parameterized and cannot be controlled to study curriculum learning with transitional problems. Therefore, we use Maia-2 (Tang et al., 2024), a state-of-the-art chess foundation model that can mimic human-level chess at various levels of competence. Maia-2 uses a *skill-aware attention* mechanism to coherently capture the spectrum of human ability, making it well suited to our study of transitional problems and curriculum learning.

**Chess Model Series.** Maia-2 takes as input a chess position along with the strength levels of both the active player and their opponent, and outputs a probability distribution over all legal moves available to the active player. The model is trained to mimic players within the following Elo ranges: $M_0 \equiv [0, 1100), M_1 \equiv [1100, 1200), \ldots, M_{10} \equiv [2000, \infty)$. We define the strength function as as the lower bound of each rating range (restricting to $[1000, 2000]$ at the extremes): $s(M_i) = 1000 + 100i$, giving us $s_0 = 1000, s_1 = 1100, \ldots, s_{10} = 2000$. We set both the active and opponent strengths to $s_i$ to ensure a consistent skill representation. Note each $M_i$ shares the same underlying architecture and parameters, with its policy differing only due to the strength input $s_i$. The policy output for a given chess position $b$ conditioned $s_i$ is

$$\pi(\mathbf{a}|b, s_i) = f_\theta(b, s_i), \tag{2}$$

with $\pi$ denoting the probability distribution over legal moves (i.e., actions) $\mathbf{a}$, and $\theta$ representing Maia-2's parameters. Each $M_i$ can be fine-tuned with the following loss function:

$$\mathcal{L}(\theta) = -\mathbb{E}_{b \sim \mathcal{D}}[\log \pi(a^*|b, s_i)] \tag{3}$$

where $\mathbb{E}_{b \sim \mathcal{D}}[\cdot]$ denotes the expectation over chess position $b$ sampled from the data distribution $\mathcal{D}$, and $a^*$ is the ground truth best move under $b$.

**Finding Transitional Chess Problems.** We determine move correctness as:

$$\phi_p(M_i) = \begin{cases} 1 & \text{if } \arg\max_{\mathbf{a}} \pi(\mathbf{a}|b, s_i) = a^* \\ 0 & \text{otherwise} \end{cases} \tag{4}$$

This allows us to map each problem $p = \{b, a^*\}$ to its transition point $\tau_p \in \{1, 2, \ldots, 10\}$ in the model series, where positions solvable or unsolvable by all models are excluded. We define transitional problems in chess from two sources. First, we define the set of **transitional positions** from regular chess games at transition point $\tau$ as:

$$\mathcal{D}_\tau^{pos} = \mathcal{D}^{pos} \mid [(\forall i < \tau, \phi_p(M_i) = 0) \wedge (\forall i \geq \tau, \phi_p(M_i) = 1)]_{p \sim \mathcal{D}^{pos}} \tag{5}$$

where $\mathcal{D}^{pos}$ represents positions sampled from human chess games from the Lichess Database[1] annotated with the best possible move according to Stockfish (Romstad et al., 2023). While regular game positions provide diverse training data, *chess puzzles* are particularly high-quality learning material for skill acquisition. Chess puzzles form a carefully crafted subset of regular positions with a unique best move that typically leads to a decisive advantage, specifically designed to isolate and teach critical tactical patterns and strategic concepts. We therefore define a second set of transitional problems for chess, the set of **transitional puzzles** at point $\tau$:

$$\mathcal{D}_\tau^{puz} = \mathcal{D}^{puz} \mid [(\forall i < \tau, \phi_p(M_i) = 0) \wedge (\forall i \geq \tau, \phi_p(M_i) = 1)]_{p \sim \mathcal{D}^{puz}} \tag{6}$$

where $\mathcal{D}^{puz}$ is a set of randomly selected chess puzzles from Lichess[2]. These transitional puzzle sets enable targeted training on the exact skills and concepts needed to progress from $s_{\tau-1}$ to $s_\tau$. We highlight that the transition point $\tau_p$ of a position $p$ (if any) is not an intrinsic property of that position, but is specific to our $\mathcal{M}$. Positions that transition at $\tau$ are precisely those that require the competence level of model $M_\tau$ to solve. Our hypothesis is that training $M_{\tau-1}$ on $\mathcal{D}_\tau$ efficiently bridges the gap in skills between $M_{\tau-1}$ and $M_\tau$. Inductively, an easy-to-hard curriculum over transitional problems should enable the rapid progression of $M_{\tau-1}$ to higher and higher levels of competence, which is challenging to do with subjective measures of difficulty.

---

[1]https://database.lichess.org/#evals

[2]https://database.lichess.org/#puzzles

## 2.3 TRANSITIONAL PROBLEMS IN MATH

**Mathematics as a Model Domain.** The benefit of using chess as a learning domain lies in the ability to perform *extrinsic* evaluations of model competence. While ratings similar to the Elo system have been applied to competitive coding and mathematics (Quan et al., 2025), there is no publicly available dataset of problems ranked by such a scoring system or series of otherwise identical models with measurably varying competence on a particular reasoning task. However, mathematical reasoning is at the frontier of research efforts in ML, and curricula have shown some success in this domain (Section 4). Therefore, we conduct an exploratory evaluation of transitional problems for mathematical reasoning, relying on an *intrinsic* evaluation of the competence of a model as its performance on the training data distribution $\mathcal{D}$. As shown in Figure 2 (left), we train a base model $M_0$ on $N$ examples drawn from $\mathcal{D}$, collecting periodic checkpoints $C_1', C_2', \ldots, C_s'$. We select as our model series a subset of these checkpoints $\mathcal{C} = \{C_1, C_2, \ldots, C_r\}$ based on validation accuracy, such that $\mathrm{performance}(C_{i+1}) - \mathrm{performance}(C_i) \geq \delta$ for every $i \in [r-1]$ and some $\delta < 1$. That is, each of the $r$ checkpoints selected differs in strength from every other model by at least $\delta$.

**Learning Model and Dataset.** As our base learner, we choose the Qwen3-0.6B-Base (Yang et al., 2025), the state of the art pretrained 'small' LLM for mathematical reasoning with 0.6 billion parameters. This model is highly capable at simpler mathematical reasoning tasks (even outperforming many larger LLMs), and improves dramatically when fine-tuned on a reasoning dataset. We study reasoning abilities on the GSM8k Cobbe et al. (2021) dataset, which consists of mathematics word problems with well-formatted answers that involve 2-10 steps of grade-school-level arithmetic. Training Qwen3-0.6B-Base on GSM8k improves its performance from 3% to over 40% when using Chain-of-Thought reasoning (Wei et al., 2022). This setup enables us to collect checkpoints spanning a wide range of competence levels while ensuring that the highest level of competence does not saturate in perfor-

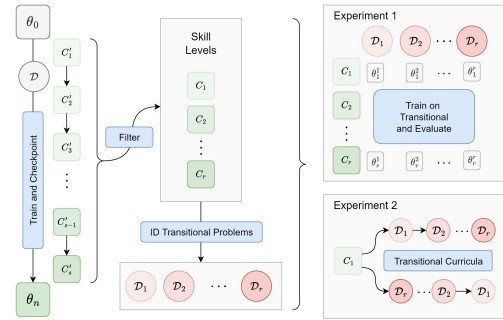

Figure 2: Pipeline for determining competence levels, identifying transitional problems, and testing leveling-up and curricula based on these problems. This pipeline applies to any trained model; we investigate a math model here.

mance on our training dataset. Thus, our experiments are able to evaluate improvement beyond the maximum performance achieved by training i.i.d. on the GSM8k dataset.

## 3 EXPERIMENTS

To study whether transitional problems can benefit model training, we train each model $M_i$ on every level of transitional problem (i.e., $\tau_1, \ldots, \tau_r$) and evaluate its performance on a held-out set of transitional problems at the next level ($\tau_{i+1}$). In the chess setting, we train on *transitional puzzles* and evaluate on *transitional positions*. This mimics how humans improve, as the high-quality puzzles are especially designed for learning. In the math setting, we evaluate on the held-out GSM8k `test` split to measure the overall improvement of the model. To empirically evaluate the benefits of curriculum learning on transitional problems, we train models with ascending (easy-to-hard), random (i.i.d.), and descending curricula, and assess their performance on the above evaluation sets, represented in Figure 2. Detailed training settings are presented in Appendix C.

### 3.1 TRAINING ON TRANSITIONAL PROBLEMS

**Chess Domain.** We fine-tune models $\{M_0, M_2, M_4, M_6, M_8\}$ on their 'level-up' transitional puzzles $\mathcal{D}_\tau^{puz}, \tau \in \{1, 3, 5, 7, 9\}$ and test on the equivalent transitional positions $\mathcal{D}_{i+1}^{pos}$ for model $M_i$. We set a fixed training budget for every setting for a fair comparison. As the results show in Figure 3, transitional problems one level up from the model's competence consistently result in the highest performance improvement. For example, the second sub-figure shows that the 1200-level ($\mathcal{M}_2$)

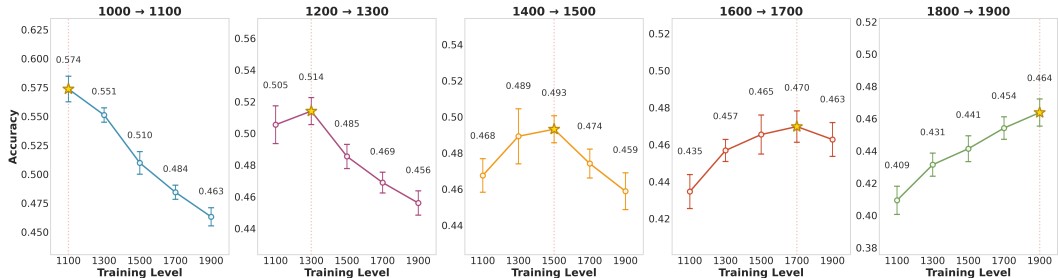

Figure 3: Each subplot depicts results when a **chess** model at a particular competence level $i$ is trained on puzzles from a single level (along the x-axis), and tested on its ability to solve game-position problems at level $i + 1$ (accuracies, y-axis). The vertical line in each indicates our hypothesized level to achieve the best performance. The star denotes the actual level that achieves the best performance, which consistently aligns with our hypothesis. Error bars represent *std* across 10 runs.

achieves the best performance on 1300-level testing problems ($\mathcal{D}_3^{pos}$) by training with 1300-level problems ($\mathcal{D}_3^{puz}$). We report the results with $\mathcal{D}^{puz}$-$\mathcal{D}^{pos}$ pair ($\mathcal{D}^{puz}$ for training, $\mathcal{D}^{pos}$ for testing) here for its similarity to human learning and the inherent difficulty in generalization to a new distribution of positions, while we observe consistent patterns with $\mathcal{D}^{puz}$ - $\mathcal{D}^{puz}$ and $\mathcal{D}^{pos}$ - $\mathcal{D}^{pos}$ pairs. Detailed results are presented in Appendix 10 and 11.

**Math Domain.** We fine-tune the Qwen2.5-0.5B-Base model ($M_0$) on the GSM8k dataset for 100 steps, checkpointing every 5 steps to collect $\mathcal{C}' = \{C_5, C_{10}, \ldots, C_{100}\}$. We select $r = 4$ checkpoints to construct our set $\mathcal{C}$ of models that characterize levels of competence. Each checkpoint in $\mathcal{C}$ is $\sim 8\%$ more accurate on the GSM8k validation set than its predecessor, with test-set accuracies ranging from 5% to 20%.

Figure 4 shows that the trend we observe in the chess setting (Figure 3) largely holds true in our math setting—training $C_i$ on transitional problems that are roughly one level up from the learner results in the greatest increase in model performance. Unfortunately, unlike the chess setting, the transitional datasets are too small for the results to be devoid of stochastic noise, even with cross-validation. We consider this to be a limitation of the available data in the reasoning domain, and hope to motivate follow-on work that focuses on a larger-scale evaluation of transitional problems in reasoning. Thus, we do not evaluate the results on the level-up problems in the math setting, and rely on the test-set generalization performance as a benchmark for evaluation. Additional results (on Qwen3-0.6B-Base) can be found in Appendix B.

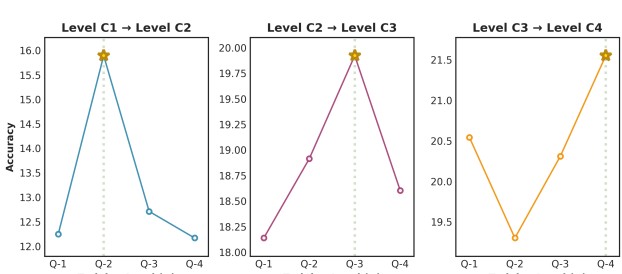

Figure 4: Performance on the held-out GSM8k test split after level-up training on transitional problems. The light green vertical line indicates our hypothesized difficulty level to achieve the best performance when training on problems from the next level. Our results corroborate this hypothesis.

### 3.2 CURRICULUM LEARNING WITH TRANSITIONAL PROBLEMS

**Chess Domain.** Training on the immediate next level of transitional problems most efficiently 'levels up' the model, which induces a natural curriculum with ascending difficulty in terms of the transitional point. We implement curricula for 2 ($\mathcal{D}_\tau, \tau \in \{1, 9\}$), 3 ($\mathcal{D}_\tau, \tau \in \{1, 5, 9\}$), and 5 ($\mathcal{D}_\tau, \tau \in \{1, 3, 5, 7, 9\}$) levels. As shown in Figure 5, IID, Asc., and Desc. denote the i.i.d baseline where training data was uniformly and randomly drawn from the combined distribution of multiple levels, e.g. IID-2 denotes data is drawn from $\{\mathcal{D}_1, \mathcal{D}_9\}$, ascending curriculum from easy to hard,

e.g., $\mathcal{D}_1$ for the first 50% training budget and $\mathcal{D}_9$ for the rest, and descending curriculum from hard to easy, e.g., $\mathcal{D}_9$ for the first 50% and $\mathcal{D}_1$ for the rest, respectively. Note that IID, Asc., and Desc. settings only differ in the order of presentation of the training samples. To ensure a fair comparison, we employ the vanilla stochastic gradient descent instead of optimizers such as Adam (Kingma, 2014) to ablate the effects of e.g., learning momentum on training samples presented at later stages.

We observe consistent wins for the ascending curriculum over the i.i.d. baseline, and consistent losses for the descending curriculum. When the training budget varies from tiny to medium (refer to Appendix 3 for details), although such patterns still exist, the win of Asc. over IID becomes less apparent. We hypothesize that IID, Asc., and Desc. will converge when a sufficient training budget is given. Notably, Figure 5 shows ascending curriculum consistently outperforms both IID and descending orderings, with the largest improvement of +3.6% observed in the $\mathcal{D}^{pos}/\mathcal{D}^{pos}$ setting. Descending curriculum shows substantial degradation, particularly in the $\mathcal{D}^{pos}/\mathcal{D}^{pos}$ setting (-10.2%). Additionally, such results hold with or without distributional shifts between puzzles and game-positions. Interestingly, we do find that both training and testing with game-positions give the most advantage to the ascending curriculum compared with other distribution pairs. Another important finding from Figure 5 and Table 1 in the appendix is that ascending curriculum learning with training and

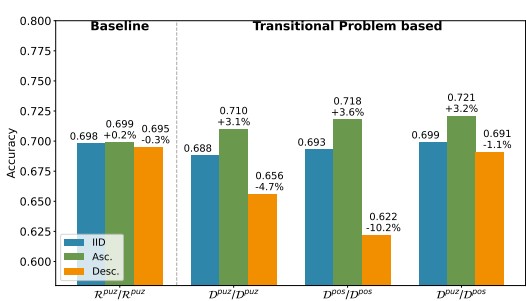

Figure 5: Performance of curriculum strategies across four **chess** evaluation settings: $\mathcal{R}^{puz}/\mathcal{R}^{puz}$ (ELO rating-based baseline), and three transitional problem-based settings, including $\mathcal{D}^{puz}/\mathcal{D}^{puz}$ and $\mathcal{D}^{pos}/\mathcal{D}^{pos}$ (in-distribution), and $\mathcal{D}^{puz}/\mathcal{D}^{pos}$ (out-of-distribution, training on puzzles and testing on positions). We evaluate three training orderings: IID (random sampling), Asc. (ascending: easy-to-hard), and Desc. (descending: hard-to-easy).

testing data leveled by ELO ratings does not outperform baselines, confirming the significance of the proposed model-centric difficulty measure based on transitional problems.

**Math Domain.** We evaluate two curriculum learning setups in the math domain. Starting from level $C_1$, we train forward, random, and reverse curricula on transitional problems from higher levels. Each curriculum consists of 3 blocks of training with a budget of 2 ('Tiny') or 5 ('Low') steps per block (Figure 7a). We find that training on the easy-to-hard curriculum outperforms the other strategies across both training budgets. We also evaluate the effectiveness of transitional problems in training the base Qwen2.5-0.5B model, and compare it with training on i.i.d. and static curricula over the GSM8k training set based on our observations in Section 3.3. We train each setup for the equivalent of 1 epoch over transitional problems (roughly 20 steps with a batch size of 64). Figure 7b shows that every curriculum on transitional problems vastly outperforms non-transitional curricula with this budget, indicating that the former problems form a highly informative subset of the full training dataset. Additionally, the easy-to-hard ordering on transitional problems outperforms other curricula, getting the base model to over 20% accuracy with just 20 steps of training.

## 3.3 INTERPRETING TRANSITIONAL PROBLEM DIFFICULTY

**Chess Domain.** Figure 6 shows that transitional chess puzzles are correlated with the puzzle rating, an extrinsic measure of puzzle difficulty. The distribution skews towards higher ratings as competence levels increase, but the mean does not increase by a large margin. This shows that the puzzle rating, while informative, is not the ideal indicator of the learning difficulty of a problem.

**Math Domain.** To evaluate whether transitional problems represent a human-interpretable easy-to-hard ordering, we measure the hardness of problems along five criteria: the problem length (#tokens); the solution length (#tokens); the distribution of reasoning steps in the solution (annotated by $<< \cdots >>$); the average arithmetic operations per problem; and the average number of operations per arithmetic skill (add/sub/mul/div) required to solve a problem. Figures 8 and 9 show that the

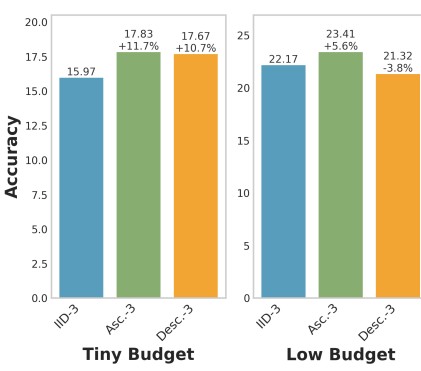 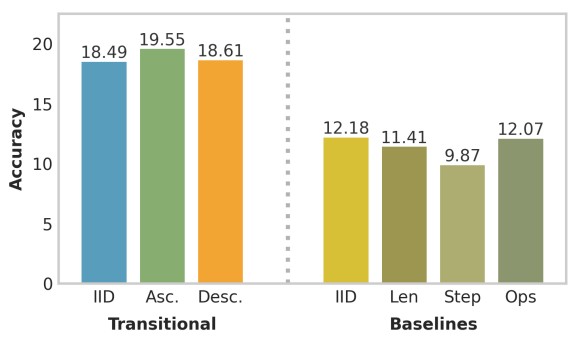

(a) Level 1 model trained on levels 2,3,4

(b) Curricula on the Qwen2.5-0.5B base model

Figure 7: Curriculum learning results in math, training Qwen2.5-0.5B on GSM8k. (a) The best training strategy to level up a Level 1 is an easy-to-hard curriculum over transitional problems. (b) 1 epoch of training on transitional problems outperforms static curriculum and i.i.d. baselines with the same amount of data.

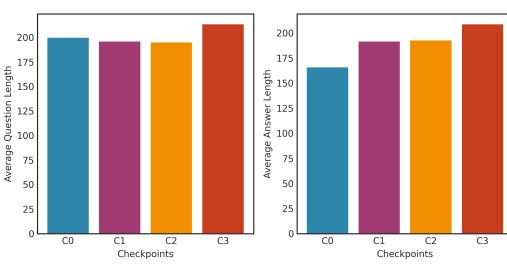 

(a) Average Question and Answer Length

(b) Average # Operations

Figure 8: Transitional problems are aligned with human notions of problem difficulty. A transitional problem at a higher competence level consists on average of a longer question and answer and more operations in the solution compared to one at a lower level.

average length and number of operations of a problem is greater for transitional problems at higher levels than at lower levels (especially evident in a stronger model such as Qwen2.5-1.5B—Appendix B). Stronger models are also able to solve problems with more GSM8k-tagged reasoning steps on average. However, the composition of skills that the problem requires does not change on average, contravening the human notion of difficulty (e.g., division being more complex than addition). Thus, transitional problems align with some intuitive notions of problem difficulty, but cannot be fully replicated simply by defining a predetermined ordering on these criteria. Notably, *examining all solved problems at a given competence level does not produce these orderings* (Appendix B), showing that our definition of transitional problems is what begets interpretability.

## 4 RELATED WORK

**Mixed Results** Despite the ubiquity of curricula for learning in humans, curriculum learning has remained a relatively niche approach for machine learning practitioners. The majority of wins for curriculum learning over training i.i.d. appear as improvements in learning efficiency and convergence time. Some curricula are defined not in terms of the difficulty of data examples but instead task difficulty. LLMs are frequently pre-trained on sequences that increase in length and context windows that grow in size over training (Liu et al., 2024). Similarly, curricula are also useful in learning sequential decision making. Reinforcement learning methods often use curricula to learn increasingly long action sequences (Narvekar et al., 2020; Patel et al., 2024; Li et al., 2025; Zhao et al., 2022) and for faster convergence (Tao et al., 2024). The training of many autoregressive foundation models has a consistent, curriculum-like structure, with large-scale pre-training phase on

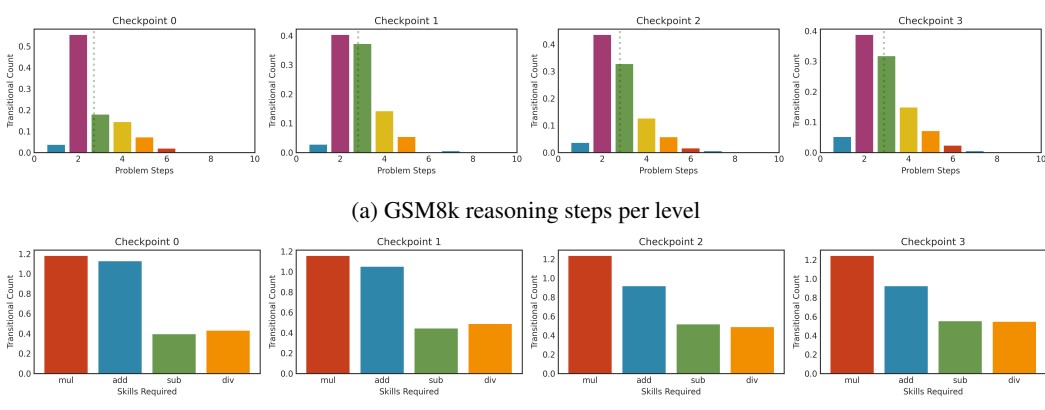

(a) GSM8k reasoning steps per level

(b) Operations per arithmetic skill across levels

Figure 9: The level of a transitional problem is mildly correlated with the number of reasoning steps in its solution, but there is no clear relation with the arithmetic skills required to solve a problem.

unstructured data followed by a structured post-training phase Brown et al. (2020). Recent training schemes include a 'mid-training' phase (Wake et al., 2024; OLMo et al., 2024) that focuses on enhancing capabilities with higher-quality data than in pre-training. The highly structured post-training phase has shown wins for curricula in training for instruction-following (Ge et al., 2025) and complex reasoning Zeng et al. (2025); Polu et al. (2022). In contrast, efforts to explicitly incorporate curriculum learning into pre-training have been largely unsuccessful, even with data inspired by human development. The BabyLM challenge (Oba et al., 2023) saw a myriad of curriculum-based approaches fail to beat the baseline for pre-training a language model on 10-100 million tokens of data. These included domain-specific (Martinez et al., 2023; Edman & Bylinina, 2023; Oba et al., 2023), model-dependent (Opper et al., 2023), and student-teacher (Chobey et al., 2023; Zhang et al., 2023) setups, but notably did not evaluate curricula that dynamically assess example difficulty to determine the next inputs during training. Other negative results for curriculum learning stem from settings where models can be trained to convergence with relatively noise-free datasets (Wu et al., 2020). One benefit of curriculum learning is that it simplifies the initial training of a model, which may not be beneficial in foundation models that are over-parameterized (Mannelli et al., 2024).

**Training Problem Difficulty** The first step in an exploration of structured training methods is to characterize training problems in a way that begets an ordering. The standard approach to is to estimate the difficulty in learning a training problem, so that easier problems may be learned by weaker models in a curriculum. The majority of existing works use *model-independent* approaches that rely on some *domain-specific* measure of the innate complexity of the data, such as the quality of an image (Wang et al., 2023; Sheybani et al., 2023), complexity of a reasoning problem (Polu et al., 2022), or the size and quality of textual data (Zhang et al., 2018; Warstadt et al., 2023). However, human-derived measures of complexity do not always correlate with the difficulty in learning an example by gradient descent, frequently leading to subpar results (Oba et al., 2023). Alternately, problem difficulty can

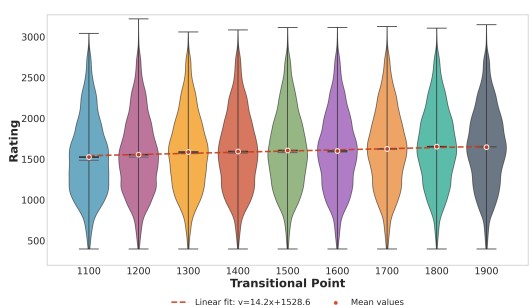

Figure 6: Rating–Transitional Point correlation in puzzles. The human-centric puzzle ratings are weakly correlated to transitional points, which shows that the transitional problems are capturing another dimension of puzzle difficulty, in particular with regard to the model competence.

be measured by leveraging the *training statistics* of a model. *Dynamic* approaches to model-based difficulty leverage per-step training information such as the reduction in training loss, gradient norm, or the complexity of model parameters on a training problem (Bellemare et al., 2016; Graves et al.,

2017). While these approaches avoid the pitfalls of model-agnostic curricula, they are sensitive to training noise and add computational overhead to the training process. *Static* approaches (like our method) use information aggregated over training runs to smooth out the effects of training stochasticity. The most similar metric to ours in the literature is the consistency score (C-score) (Jiang et al., 2020), which aggregates the transfer learning performance of models trained on diverse pretraining sets on a problem from the target training set. While each pretrained model differs in competence, there isn't necessarily a clear hierarchy of models for easy-to-hard ordering of training samples by C-score. Experiments that leverage the C-score in curriculum learning achieve mixed results, only achieving wins for curriculum learning on noisy or limited data (Wu et al., 2020). Our definition of transitional problems provides a more clear path towards curriculum learning.

**Structured Training** Given a strategy for ordering examples by learning difficulty, the natural question is to identify the best way to leverage this meta-information to improve training. The two major approaches to structured training are *active learning* (Cohn et al., 1994) and *curriculum learning*. Active learning optimizes for the *short-term* goal of finding the best next problem(s) to train a learning model on. Strategies for active learning generally rely on auxiliary 'teacher models' to select batches that attempt to minimize the uncertainty over the training data distribution, using methods such as data clustering (Citovsky et al., 2021), influence functions (Liu et al., 2021), and coresets (Sener & Savarese, 2017) to identify suitable problems. These methods tend to add considerable overhead and require strong estimates of learner uncertainty, and are thus not popular for training massive-scale foundation models. Contrary to this approach, curriculum learning optimizes for the *long-term* goal of training a model on a sequence of batches, focusing on learning performance at the end of the training phase instead of per-step uncertainty. Some methods combine these approaches to develop curricula with dynamic batch selection using teacher-student models (Matiisen et al., 2019) or multi-armed bandits (Graves et al., 2017). We focus on developing a static curriculum over transitional problems with the aim of avoiding the costly computations of dynamic curricula while identifying a small set of problems that can make efficient learning progress.

**Human Learning** Our strategy for 'leveling up' ML models with transitional problems is strongly motivated by the structure of curriculum-driven human learning. A learning regimen for humans typically emphasizes sub-tasks that are just out of reach of a learner's current capabilities. In psychology, the technique of *scaffolding* (Wood et al., 1976) is used to teach infants new skills (such as object detection and manipulation) by focusing on tasks in their *Zone of Proximal Development* (ZPD) (Vygotsky, 1978), i.e., tasks that can be accomplished with some assistance from a teacher. Scaffolding is analogous to the concept of training on the transitional problems of the next level in our work. Neural networks trained on egocentric video data from infants perform best when trained on a developmental (young-to-old) ordering (Sheybani et al., 2023). Beyond infancy, ordering concepts by complexity is the standard in the scholastic system, and has even been shown to help adults in motor learning tasks (Sungeelee et al., 2024).

## 5 DISCUSSION

In this work, we introduce the notion of *transitional problems* by identifying the training problems that uniquely define the level of competence of an ML model, and are also well-calibrated to models across every level of competence. This enables a true easy-to-hard ordering of problems relative to the levels of competence that a model is actually capable of achieving. Through experiments across chess and mathematics, we observe that: (1) training on transitional problems from the next level of competence enables models to efficiently progress towards "leveling up" in performance; and (2) curriculum learning with transitional problems outperforms other strategies, indicating that our method produces a useful easy-to-hard ordering of training problems.

A limitation of our work is that we primarily explore static curricula in this work, and follow-on research could design and evaluate dynamically selected transitional problems, adapted to the individual learner. Additionally, transitional problems are inherently defined with respect to a model at a higher level, necessitating that such a model be available prior to training. Future work could aim to remove this dependency by, e.g., predicting a learner's level-up problems based on various levels of competence observed in a diverse *model classroom* or use transitional problems to improve the model at the highest level.

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

# A ADDITIONAL CHESS RESULTS

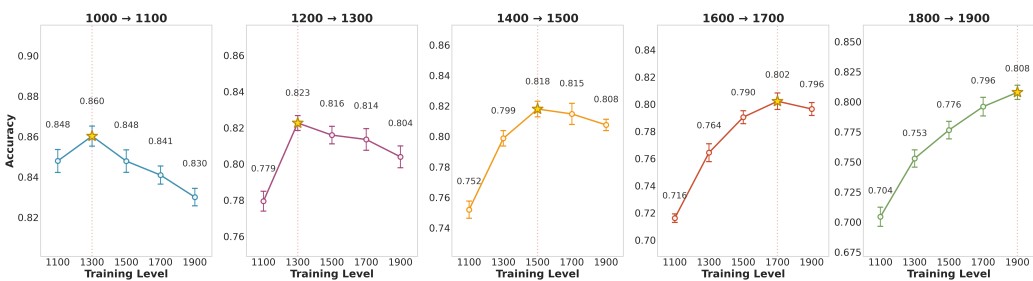

Figure 10: Training on various levels of puzzles and testing on one level up puzzles w.r.t the competence of the model to be fine-tuned. The vertical line indicates our hypothesized level to achieve the best performance. The star denotes the actual level that achieves the best performance, which consistently aligns with our hypothesis. Error bars represent *std* across 10 runs.

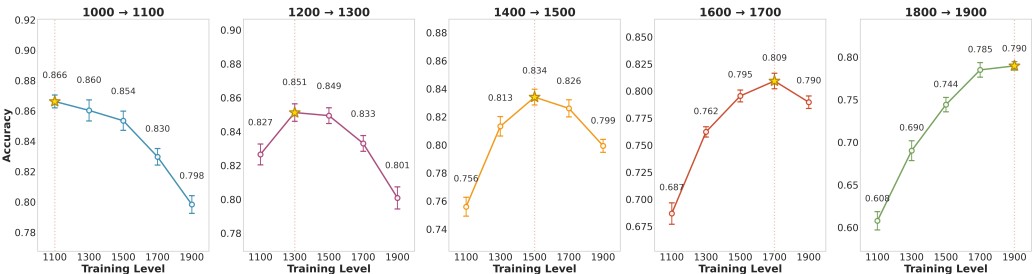

Figure 11: Training on various levels of game-positions and testing on one level up game-positions w.r.t the competence of the model to be fine-tuned. The vertical line indicates our hypothesized level to achieve the best performance. The star denotes the actual level that achieves the best performance, which consistently aligns with our hypothesis. Error bars represent *std* across 10 runs.

| | Baseline | | Transitional Problem based Difficulty Definition | | | | | |
|---|---|---|---|---|---|---|---|---|
| | | | In-Distribution | | | | Out-of-Distribution | |
| | $\mathcal{R}^{puz}/\mathcal{R}^{puz}$ | Impr | $\mathcal{D}^{puz}/\mathcal{D}^{puz}$ | Impr | $\mathcal{D}^{pos}/\mathcal{D}^{pos}$ | Impr | $\mathcal{D}^{puz}/\mathcal{D}^{pos}$ | Impr |
| IID-2 | 0.692 | - | 0.680 | - | 0.677 | - | 0.696 | - |
| Asc.-2 | 0.693 | +0.1% | 0.698 | +2.6% | 0.698 | +3.0% | 0.717 | +3.0% |
| Desc.-2 | 0.690 | -0.3% | 0.665 | -2.2% | 0.603 | -11.1% | 0.684 | -1.7% |
| IID-3 | 0.698 | - | 0.688 | - | 0.693 | - | 0.699 | - |
| Asc.-3 | 0.699 | +0.2% | 0.710 | +3.1% | 0.718 | +3.6% | 0.721 | +3.2% |
| Desc.-3 | 0.695 | -0.3% | 0.656 | -4.7% | 0.622 | -10.2% | 0.691 | -1.1% |
| IID-5 | 0.701 | - | 0.690 | - | 0.694 | - | 0.707 | - |
| Asc.-5 | 0.699 | -0.3% | 0.697 | +1.0% | 0.716 | +3.1% | 0.721 | +2.0% |
| Desc.-5 | 0.695 | -0.8% | 0.663 | -3.8% | 0.642 | -7.5% | 0.696 | -1.5% |

Table 1: Comparison of curriculum learning with 2, 3, and 5 difficulty levels in Chess. $\mathcal{D}^{puz}/\mathcal{D}^{pos}$ denotes training on transitional puzzles, testing on transitional positions. $\mathcal{R}^{puz}/\mathcal{R}^{puz}$ denotes training and testing on puzzles leveled by ELO ratings as the difficulty measure. Percentages indicate relative performance compared to the IID baseline.

In particular, Figure 12 shows that the ascending curriculum improved the IID baseline by at most 31.4% (Asc.-2 vs IID-2 with Medium training budget on the right subfigure), demonstrating the effectiveness of curriculum learning from easy to hard with our defined transitional problem-based difficulty.

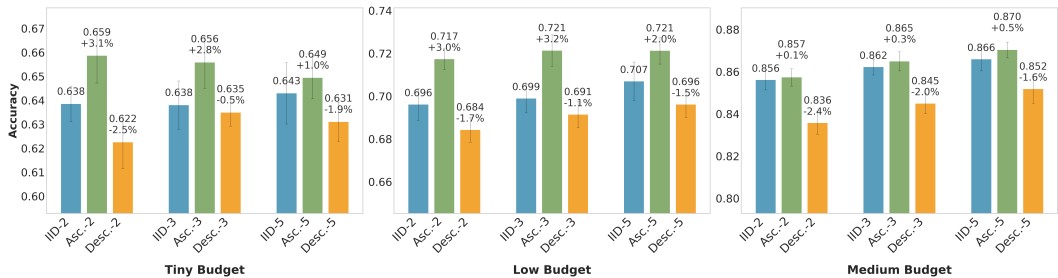

Figure 12: Performance comparison of curriculum learning strategies for chess models across different computational budgets and number of included levels. Percentages indicate relative performance compared to the IID baseline.

**Additional Observation.** As shown in Figure 12, we did not find notable patterns in the effect of curriculum steps in chess. However, Figure 6 shows that chaining up with 5 levels consistently outperforms 2 levels in math, suggesting that more curriculum steps may better help model learning in math. We would like to clarify that our goal to include multiple settings in terms of steps is to show the *consistency* of the advantage of ascending curricula over baselines, instead of finding the best hyperparameter setting.

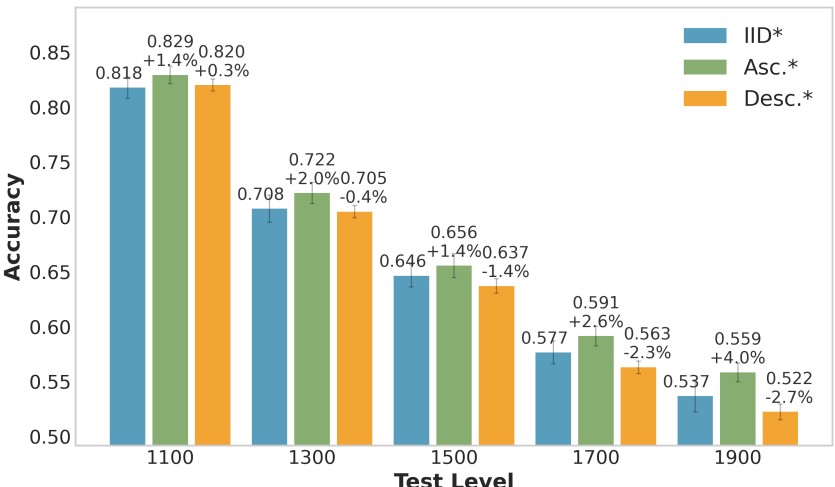

Figure 13: Performance comparison of curriculum learning strategies across different transition points under **"tiny"** training budget. IID*, Asc.*, and Desc.* denote the best performing i.i.d baseline, ascending curriculum from easy to hard, and descending curriculum from hard to easy, respectively. The best performing training strategy is selected among using 2, 3, or 5 levels. Percentages indicate relative performance compared to the IID* baseline. Error bars represent standard deviation across 10 runs with randomly split training and testing sets.

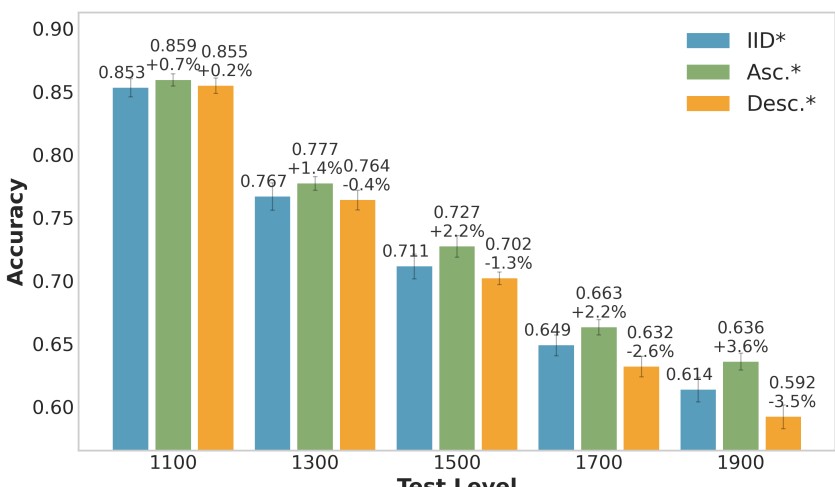

Figure 14: Performance comparison of curriculum learning strategies across different transition points under **"low"** training budget. IID*, Asc.*, and Desc.* denote the best performing i.i.d baseline, ascending curriculum from easy to hard, and descending curriculum from hard to easy, respectively. The best performing training strategy is selected among using 2, 3, or 5 levels. Percentages indicate relative performance compared to the IID* baseline. Error bars represent standard deviation across 10 runs with randomly split training and testing sets.

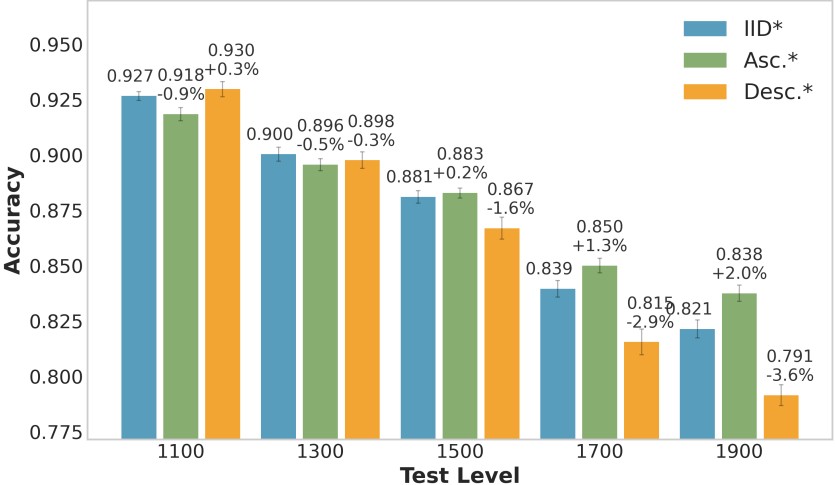

Figure 15: Performance comparison of curriculum learning strategies across different transition points under **"medium"** training budget. IID*, Asc.*, and Desc.* denote the best performing i.i.d baseline, ascending curriculum from easy to hard, and descending curriculum from hard to easy, respectively. The best performing training strategy is selected among using 2, 3, or 5 levels. Percentages indicate relative performance compared to the IID* baseline. Error bars represent standard deviation across 10 runs with randomly split training and testing sets.

## B  ADDITIONAL MATH RESULTS

### B.1  TRANSFER LEARNING ON TRANSITIONAL PROBLEMS

As defined, transitional problems rely on a pre-existing model series evaluated on a training dataset. While these models need not necessarily be checkpoints from training our target model, this restriction still limits the ability to apply transitional-based curricula on new datasets. A potential solution to this is to *find analogous transitional problems to the target training dataset in a related reference dataset*. With access to a (potentially much smaller) dataset in a similar concept domain as our target dataset, we can use existing measures of problem similarity to identify *neo-transitional* problems on our target dataset, as the sets of examples that are most similar to each level of transitional training problems from our reference dataset.

As a proof of concept, we evaluate the transfer of transitional problems from the GSM8k dataset to Orca (Mitra et al., 2024), a large-scale pretraining dataset of grade-school mathematics word problems designed specifically to improve performance on GSM8k. Meant for pretraining, Orca contains poorly formatted solutions; we use the DeepSeek-R1 distilled Qwen3-8B model (DeepSeek-AI et al., 2025) to format the solutions of 50,000 Orca problems into the style of GSM8k for better problem matching, with filters to ensure consistency with the new format and the original final answer. Our `orca-gsm8k-formatted` dataset consists of around 6000 training problems, of which 3800 are training problems and 2554 are test problems. We then use the Qwen3-Embedding-8B model (Zhang et al., 2025) to embed the problems and solutions of GSM8k and `orca-gsm8k-formatted` into vectors, specifying in the prompt that the similarity between problems should be maximized when they require the same amount of effort to solve. From this, we compute the cosine similarity between GSM8k transitional problems and Orca problems to create the *neo-transitional* problems on `orca-gsm8k-formatted`. Analogously to Figure 7b, compare training one epoch on neo-transitional problems to training for the same number of compute steps on various curricula over problems sampled i.i.d. from `orca-gsm8k-formatted`.

Table 2: The performance of various curricula for training Qwen2.5-0.5B on *neo-transitional problems* on a formatted version of the Orca dataset. Despite being formed by analogy from transitional problems on a related dataset (GSM8k), these problems largely show the same benefits as training on transitional problem derived by evaluating a model series.

| #Samples | Batch Size | Neo-Transitional | | | Full Dataset | | | |
|---|---|---|---|---|---|---|---|---|
| | | IID | Asc. | Desc. | Length | Steps | Ops | IID |
| 581 | 64 | 21.6 | **23.2** | 14.6 | 22.2 | 16.1 | 16.8 | 16.2 |

Table 2 shows that the easy-to-hard curriculum outperforms every other training method on both neo-transitional and random training problems. Training on randomly ordered transitional problems considerably outperforms every setting on randomly sampled problems from the dataset, with the exception of the length-ordered curriculum. The high performance of the length-ordered curriculum is likely due to a quirk of the model-derived dataset formatting. Overall, this experiment shows that transitional problems are useful as seeds for curricula on other datasets, potentially solving the 'chicken-and-egg' problem of defining a model series on a training dataset for a similar model.

### B.2  ADDITIONAL EXPERIMENTS

**Results on Additional Models.** We fine-tune the Qwen3-0.6B-Base model ($M_0$) on the GSM8k dataset for 100 steps, checkpointing every 5 steps to collect $\mathcal{C}' = \{C_5, C_{10}, \ldots, C_{100}\}$. We select $r = 7$ checkpoints to construct our set $\mathcal{C}$ of models that characterize levels of competence. Each checkpoint in $\mathcal{C}$ is $\sim 5\%$ more accurate on the GSM8k validation set than its predecessor. We observe that the number of transitional problems increases from 30 to 500 as the competence level of the model increases. Since even the largest of these sets is relatively small for LLM fine-tuning, we conduct *5-fold cross-validation* to verify the robustness of our findings.

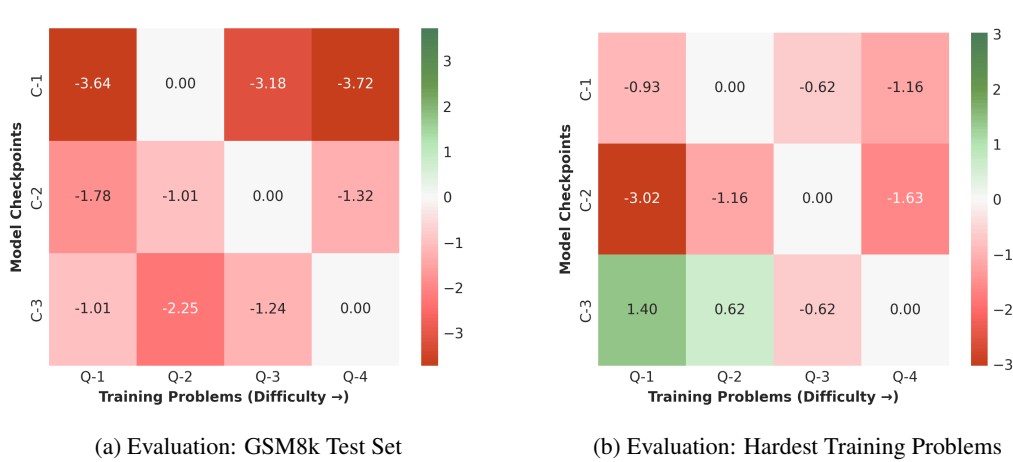

(a) Evaluation: GSM8k Test Set

(b) Evaluation: Hardest Training Problems

Figure 16: **Qwen2.5-0.5B on GSM8k**: Performance of models trained on transitional problems relative to training on transitional problems from the next competence level, evaluated on (a) problems from the held-out GSM8k test set; (b) problems that were unsolved by all models prior to transitional training.

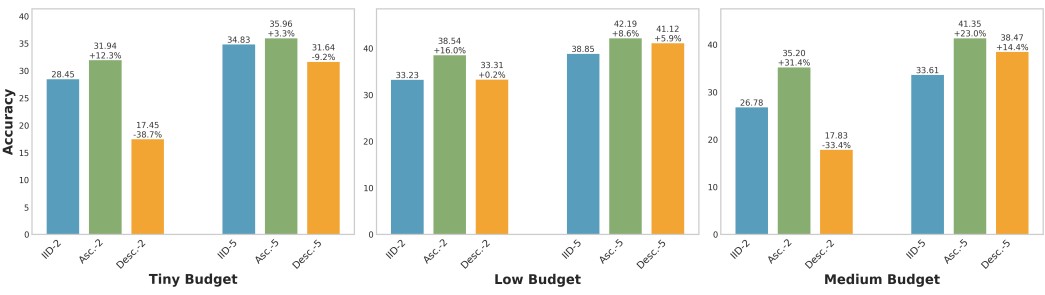

Figure 17: Performance comparison of curriculum learning strategies for math models across different computational budgets and number of included levels. IID, Asc, and Desc respectively denote the i.i.d baseline, easy-to-hard curriculum, and reverse curriculum over 2 or 5 training blocks. Percentages indicate relative performance compared to the IID baseline.

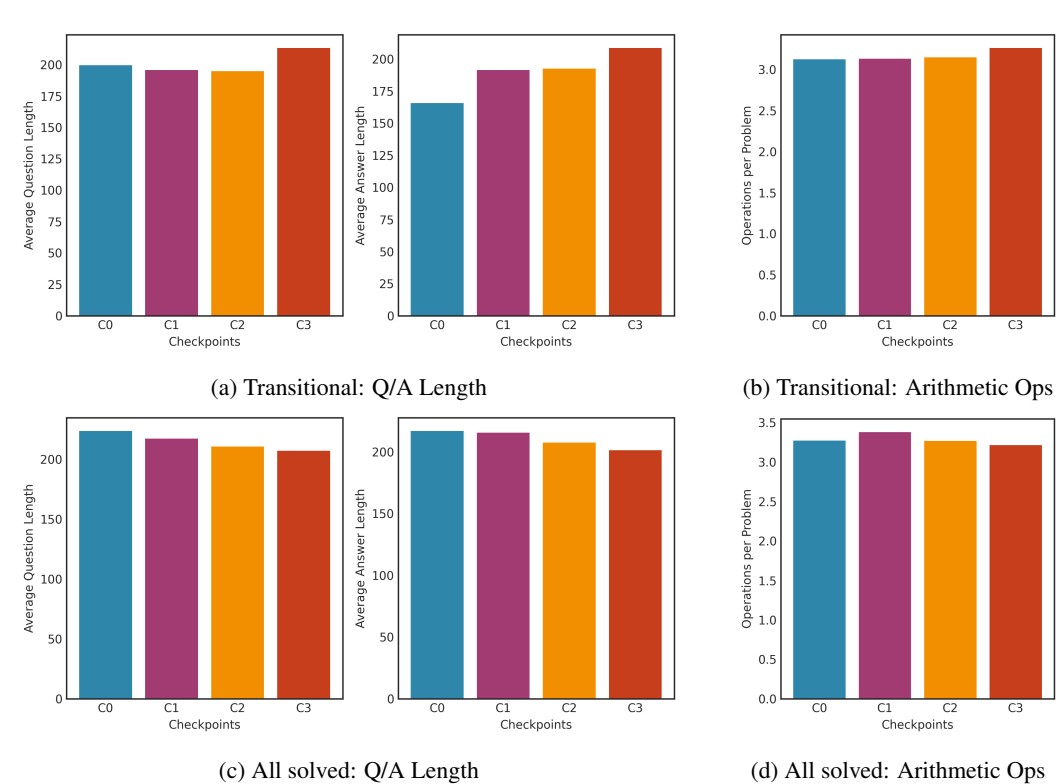

(a) Transitional: Q/A Length

(b) Transitional: Arithmetic Ops

(c) All solved: Q/A Length

(d) All solved: Arithmetic Ops

Figure 18: **Qwen2.5-0.5B on GSM8k**: Transitional problems are aligned with human notions of problem difficulty. A transitional problem corresponding to a higher competence level consists on average of a longer question and answer, and more operations in the solution, compared to a transitional problem corresponding to a lower level. The same pattern is not visible by simply looking at these statistics for the set of all problems solved by a model at a given competence level.

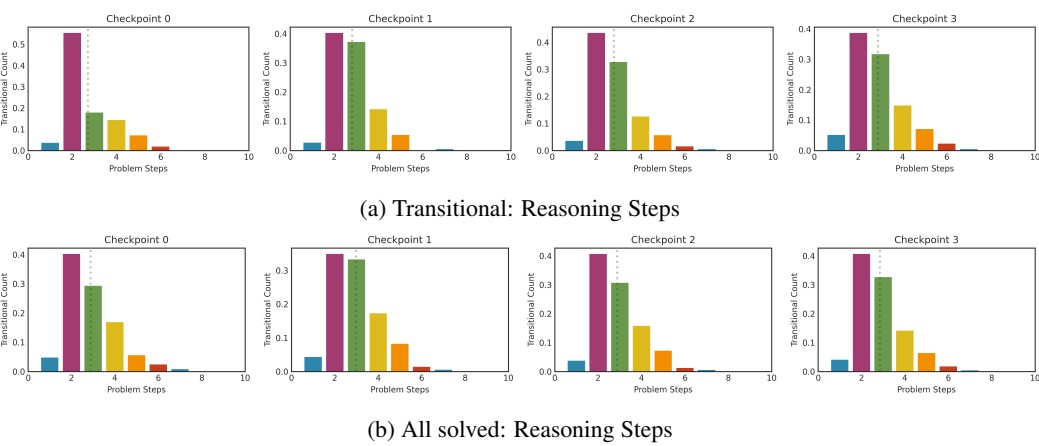

(a) Transitional: Reasoning Steps

(b) All solved: Reasoning Steps

Figure 19: **Qwen2.5-0.5B on GSM8k**: Transitional problems at higher levels require more steps to solve on average than transitional problems at lower levels. The toughest problems that strong models are able to solve problems also require more reasoning steps than those that weaker models can solve.

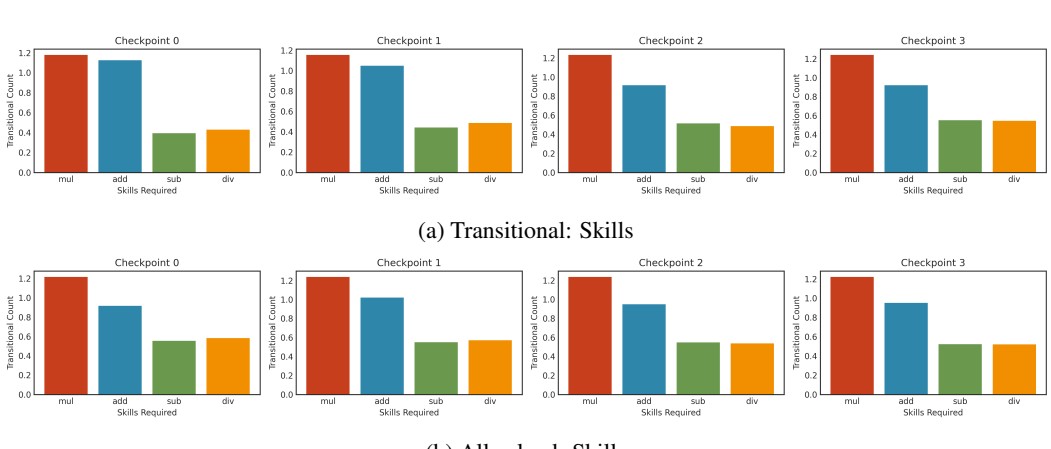

(a) Transitional: Skills

(b) All solved: Skills

Figure 20: **Qwen2.5-0.5B on GSM8k**: Contrary to the human-oriented notion of difficulty (e.g., division being more complex than addition), there is no significant trend in the composition of skills required to solve easier vs. harder transitional problems. This shows that transitional problems capture a more nuanced measure of difficulty than simple human-interpretable measures.

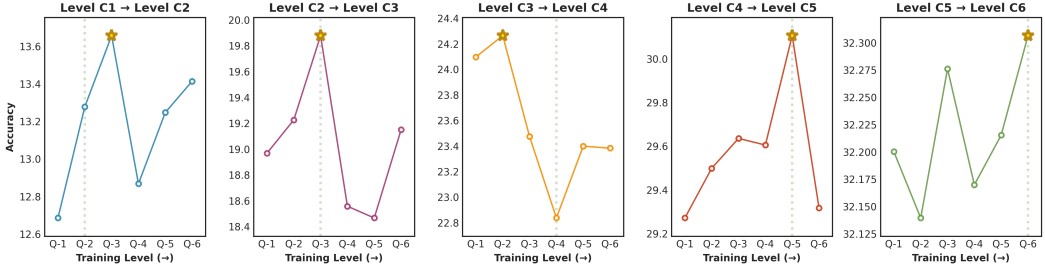

Figure 21: **Qwen3-0.6B-Base on GSM8k** Performance on the common held-out **math** split after 5 steps of level-up training. The green vertical line indicates our hypothesized best performance, when training on problems from the next level. Our results largely corroborate this hypothesis.

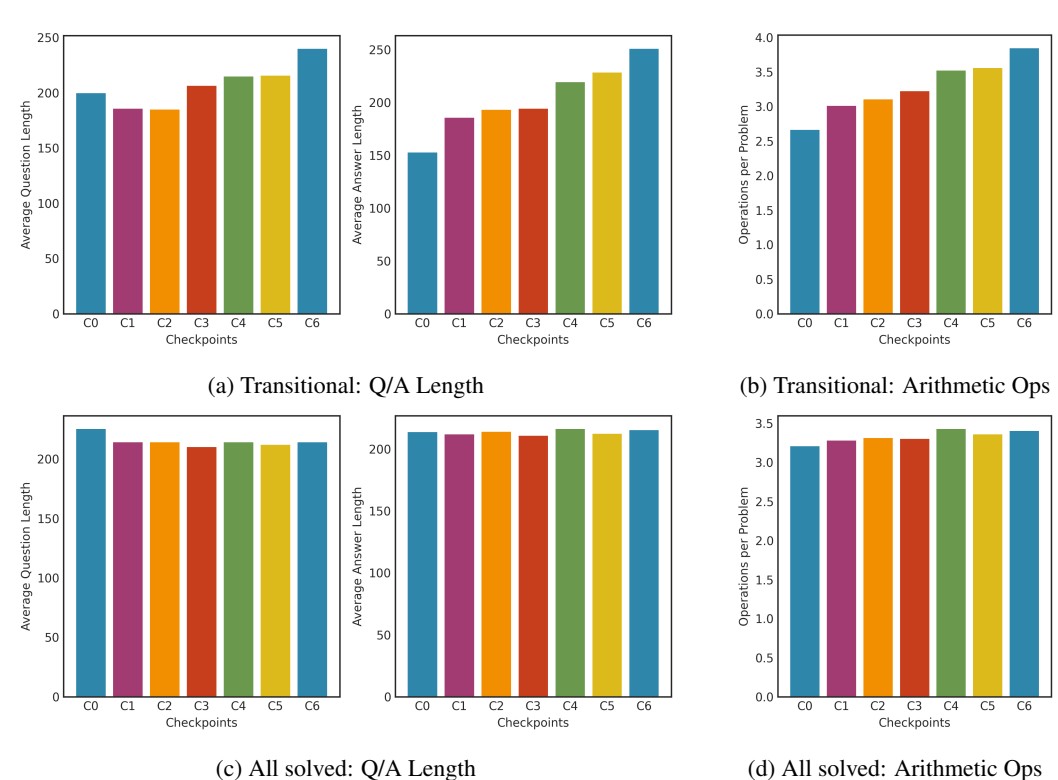

(a) Transitional: Q/A Length        (b) Transitional: Arithmetic Ops

(c) All solved: Q/A Length        (d) All solved: Arithmetic Ops

Figure 22: **Qwen2.5-1.5B on GSM8k**: Transitional problems are aligned with human notions of problem difficulty. A transitional problem corresponding to a higher competence level consists on average of a longer question and answer, and more operations in the solution, compared to a transitional problem corresponding to a lower level. The same pattern is not visible by simply looking at these statistics for the set of all problems solved by a model at a given competence level.

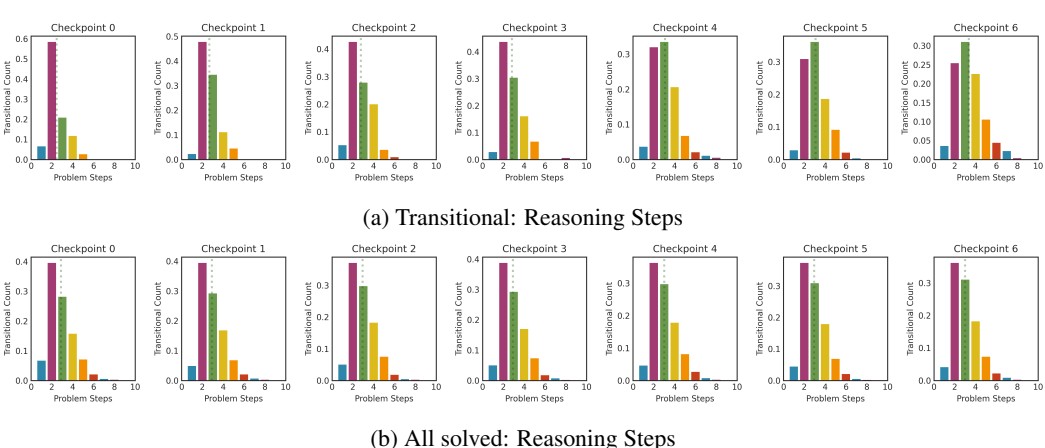

(a) Transitional: Reasoning Steps

(b) All solved: Reasoning Steps

Figure 23: **Qwen2.5-1.5B on GSM8k**: Transitional problems at higher levels require more steps to solve on average than transitional problems at lower levels. The toughest problems that strong models are able to solve problems also require more reasoning steps than those that weaker models can solve.

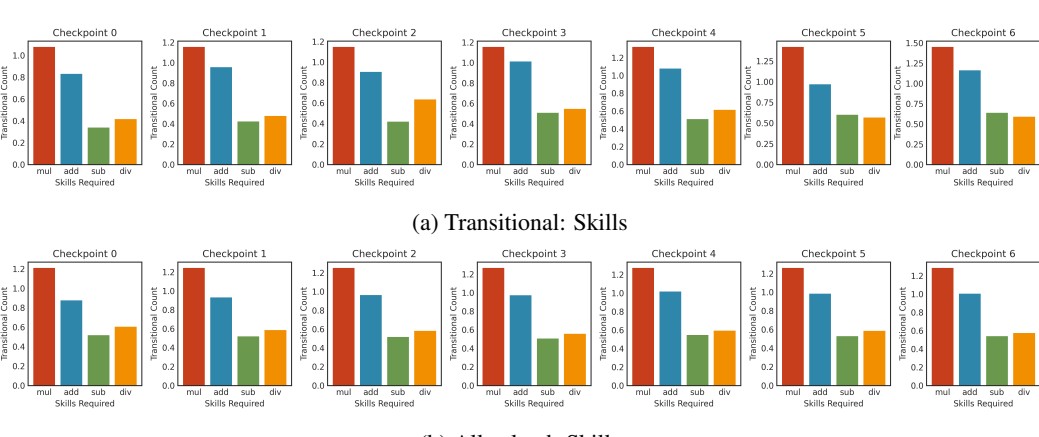

(a) Transitional: Skills

(b) All solved: Skills

Figure 24: **Qwen2.5-1.5B on GSM8k**: Contrary to the human-oriented notion of difficulty (e.g., division being more complex than addition), there is no significant trend in the composition of skills required to solve easier vs. harder transitional problems. This shows that transitional problems capture a more nuanced measure of difficulty than simple human-interpretable measures.

## C REPRODUCIBILITY

In this section, we describe our experimental settings in full. We aim to provide enough detail to independently reproduce our results from this paper along.

Table 3: Hyperparameter Settings in Chess Experiments.

| Budget | Learning Rate | Weight Decay | #Train | #Test | #Runs | Max Steps | Batch Size |
|---|---|---|---|---|---|---|---|
| Tiny | $10^{-4}$ | $10^{-6}$ | 10,000 | 10,000 | 10 | 45 | 32 |
| Low | $10^{-4}$ | $10^{-6}$ | 10,000 | 10,000 | 10 | 45 | 64 |
| Mid | $10^{-4}$ | $10^{-6}$ | 10,000 | 10,000 | 10 | 90 | 128 |

Table 4: Hyperparameter Settings in the Math Experiments.

| Stage | LR | LR Scheduler | Run Seed(s) | Max Steps | Batch Size |
|---|---|---|---|---|---|
| Get Levels | $5 \times 10^{-6}$ | inverse_sqrt | 42 | 100 | 64 |
| Transitional | $5 \times 10^{-6}$ | inverse_sqrt | 30,31,32,33,34 | 5 | 64 |
| Curr-Tiny | $5 \times 10^{-6}$ | inverse_sqrt | 101 | 10 | 64 |
| Curr-Low | $5 \times 10^{-6}$ | inverse_sqrt | 101 | 20 | 64 |
| Curr-Mid | $5 \times 10^{-6}$ | inverse_sqrt | 101 | 30 | 64 |

In the math setting, models are evaluated for their exact match accuracy to the final answer of a problem (a single number). Accuracy per question calculated as Avg@8, i.e., an average over 8 attempts. Generation was performed with the following parameters, recommended by the model developers: temperature=0.75 and top-p=0.95. The generation prompt used is similar to the one used to evaluate models in GSM-Symbolic (Mirzadeh et al., 2025).

