# OpenReview forum: "Level Up: Defining and Exploiting Transitional Problems for Curriculum Learning"
_ICLR.cc/2026/Conference — Submitted to ICLR 2026_

### Official Review · Reviewer_Ycia · 2025-10-24

**Soundness:** 4
**Presentation:** 3
**Contribution:** 3
**Rating:** 6
**Confidence:** 3

**Summary:**

The paper proposes a framework for evaluting the difficulty of data via "Transitional Problems". The authors argue that this approach is more efficient gradient-based approaches and generalized better than domain-specific curriculum. Transition problems require a set of models of different strengths, which are then monotonically ordered according to a performance metric; a sample's difficulty is then rated according to its transition point, i.e. where Model $i$ fails on said point but model $i+1$ (and all subsequent models) succeed on the sample.
The author's hypothesis is that training a model with level $t-1$ on datapoints with transition point $t$ is effectively the best data to "level-up" the model into a skill equivalent to model $t$.

The authors provide experiments in both the chess and math domain and show that (1) in Fig 3, starting from multiple model levels, the data subset that will best improve performance is not necessarily the hardest or the easiest data, but really the dataset that's "one-level up". Then, (2) the authors leverage this insight to design a curriculum training approach, where the monotonically increasing levels are presented sequentially to the learner, showing it ouperforms random sampling and reverse curriculum.

**Strengths:**

The framework presented by the authors is clear, and departs from standard approaches to assess the difficulties of samples and curriculum generation. What I especially like is the tailoring of the curriculum to the model in the math and chess settings.

The experiments are well designed. The proposed approach has potentially nice applications in the chess setting, where one could leverage this data curriculum to better train human players depending on their level.

**Weaknesses:**

1. My biggest criticism of this work lies in the "chicken-and-egg" problem to actually use this for model training. This method can design model-specific curricula from having an already trained model. To this end, what's the use of finding the optimal data ordering specific to a model if said model is already trained ?

2. The results for the curriculum learning for math are somewhat inconclusive. It seems that this approach only shows gains in the very limited dataset size regime. This is a far cry from the author's motivation tying curriculum learning to the different training phases of foundation models, which have many (many) orders of magniture more samples.

3. The claim that this appraoch is more computationally efficient than gradient based approaches is unclear. The authors still require multiple inference calls per datapoint, which is more expansive than computing a single gradient. Could the authors please provide more details for this?

**Questions:**

1. What happens in the lower-triangular part of Figure 4 and Figure 5 ? For completeness, it would be good to show that training on "levels down" leads to worse performance.

2. "At Internet scale, the potential for faster convergence with
curriculum learning methods becomes a valuable consideration due to the considerable expense of
training." -> Thi is not a Curriculum learning argument, it should be a continual learning one (cannot assume data distribution is monotonically increasing in difficulty.)

3. Related to my previous comment about the "chicken-and-egg" situation, could you provide other applications or usages of your method beyond model (re) training?

---

> ### Author Response · Authors · 2025-11-26
>
> *Chicken-and-egg problem*
>
> Please refer to General Response 1.
>
>
> *More stable math results*
>
> Please refer to General Response 2.
>
>
> *Training Efficiency*
>
> Our method requires one forward pass through each example in the training dataset to identify the transitional problems on a predefined model series. Identifying a model series similarly requires only S backward passes and C\*S forward passes through the dataset, corresponding to training the model for S steps. Typically, S >> C, i.e., checkpointing is infrequent. This fixed, one-time work produces a permanent difficulty label for every sample, with no additional cost during curriculum learning. In contrast, gradient-based curricula (loss change, gradient norm, etc.) rely on one backward pass per training sample for every training step to identify the next batch of examples, resulting in S\*S backward passes and S\*S forward passes through the example. The considerable increase in expensive backward passes shows that gradient-based curricula are significantly more expensive than a curriculum over transitional problems, which amortizes the cost of identifying these problems.
>
>
> *Relation to continual learning*
>
> We thank the reviewer for this clarification opportunity. Our claim concerns sample efficiency (fewer gradient steps to reach target performance), which is the domain of curriculum learning. Continual learning addresses a different challenge, i.e., preventing forgetting when adapting to new distributions, which is orthogonal to our argument. We have replaced the original sentence in the revision (in red font) with the following to make it clearer: “Given the considerable computational expense of large-scale training, methods that improve sample efficiency through strategic data selection and ordering become valuable considerations.”

---

### Official Review · Reviewer_zeBW · 2025-11-02

**Soundness:** 3
**Presentation:** 2
**Contribution:** 2
**Rating:** 4
**Confidence:** 2

**Summary:**

This paper introduces **transitional problems**, which are examples that weaker models fail but slightly stronger models can always solve. Instead of using human-defined difficulty or extra computation to estimate it, the authors define difficulty based on the model’s actual abilities. They test this idea in chess and math reasoning, showing that training a model on the transitional problems from just one level above its current skill leads to the fastest improvement, beating random training and reverse curricula. The core claim is that curriculum learning works when the data difficulty is aligned with the learner’s real competence. The main limitation is that finding these problems currently depends on having access to a stronger model, which the paper leaves as future work to address.

**Strengths:**

1. The paper proposes a clear and original way to define task difficulty based on the model’s actual capability, instead of relying on human judgment or handcrafted heuristics. This directly addresses a long-standing issue in curriculum learning.
2. The approach is validated in two very different domains, chess and math reasoning, and shows consistent improvements in both.
3. The experimental setup is well controlled. The comparison between ascending curriculum, i.i.d training, and reverse curriculum makes it easy to see where the gains come from.

**Weaknesses:**

1. Practical constraints in defining transitional problems
* The approach relies on the existence of a strictly stronger model in order to identify transitional problems. This requirement limits its practicality, as it introduces a non-trivial upfront cost. In effect, the method assumes that one must first perform a less efficient stage of training in order to later enable more efficient learning, which creates an inherent paradox.
2. Limited scale and stability in the math experiments
* While the results in the chess domain are highly stable, the outcomes in the math setting exhibit greater variance and weaker overall gains. This raises concerns about the generality of the proposed method, especially in reasoning-oriented tasks where dataset size and quality remain significant limitations.
* If training on the immediately higher level helps performance, it would be helpful to show how gradual transition learning compares to training with all data randomly mixed.
3. Uncertain separability of capability levels across model checkpoints
* The paper assumes clear performance gaps between successive model versions, yet this assumption may not hold in practice. In the context of LLM fine-tuning in particular, stochastic training dynamics can make it difficult to define a clean, deterministic notion of “one level higher” model capability.
4. The experiments are conducted on only one model, so it is difficult to see whether the proposed explanation applies to all models, especially large foundation models.
5. The description of the IID baseline could be made clearer.
6. The performance gain obtained through transitional learning is less than 1%, which seems very marginal, and there is concern about whether transitional problems can be defined across diverse tasks.

**Questions:**

* How does the size of the transitional problem set relate to performance gains? The paper does not include experiments that vary or reduce the amount of transitional data, so the scaling behavior remains unclear.
* Is it possible to estimate transition points without access to a stronger reference model?
* How does the length of the curriculum steps affect performance? A more detailed analysis could clarify whether there is an optimal depth.
* In the chess domain, how much do distributional differences between puzzles and positions influence the reported results?

---

> ### Author Response · Authors · 2025-11-26
>
> *Dependence on a stronger model*
>
> Please refer to General Response 1.
>
>
> *Training efficiency*
>
> Our method requires one forward pass through each example in the training dataset to identify the transitional problems on a predefined model series. Identifying a model series similarly requires only S backward passes and C\*S forward passes through the dataset, corresponding to training the model for S steps. Typically, S >> C, i.e., checkpointing is infrequent. This fixed, one-time work produces a permanent difficulty label for every sample, with no additional cost during curriculum learning. In contrast, gradient-based curricula (loss change, gradient norm, etc.) rely on one backward pass per training sample for every training step to identify the next batch of examples, resulting in S\*S backward passes and S\*S forward passes through the example. The considerable increase in expensive backward passes shows that gradient-based curricula are significantly more expensive than a curriculum over transitional problems, which amortizes the cost of identifying these problems.
>
>
> *Capability levels in LLM checkpoints* and *More models*
>
> Please refer to General Response 2.
>
>
> *IID baseline results*
>
> Relevant results can be found in Section 3.2, where we presented “how gradual transition learning compares to training with all data randomly mixed” as mentioned by the reviewer.
> Additionally, we add Figure 5 and Table 1 in the appendix to the paper to highlight the advantage of the easy-to-hard curriculum with our defined transitional problem-based difficulty, and we provide the table below for visibility, where $D^{puz}/D^{pos}$ denotes training on transitional puzzles, testing on transitional positions; $D^{pos}/D^{pos}$ denotes training on transitional positions, testing on transitional positions; and $R^{puz}/R^{puz}$ denotes training and testing on puzzles leveled by ELO ratings as the difficulty measure. This table shows that chaining up the level-up training strategy with an ascending curriculum consistently outperforms baselines, including the concerned IID setting. We have improved Section 3.2 accordingly (marked in red).
>
> |          | $R^{puz}/R^{puz}$ | Impr   | $D^{puz}/D^{puz}$ | Impr   | $D^{pos}/D^{pos}$ | Impr    | $D^{puz}/D^{pos}$ | Impr   |
> |----------|:-----------:|:------:|:-----------:|:------:|:-----------:|:-------:|:-----------:|:------:|
> | IID-2    | 0.692       | -      | 0.680       | -      | 0.677       | -       | 0.696       | -      |
> | Asc.-2   | 0.693       | +0.1%  | 0.698       | +2.6%  | 0.698       | +3.0%   | 0.717       | +3.0%  |
> | Desc.-2  | 0.690       | -0.3%  | 0.665       | -2.2%  | 0.603       | -11.1%  | 0.684       | -1.7%  |
> | IID-3    | 0.698       | -      | 0.688       | -      | 0.693       | -       | 0.699       | -      |
> | Asc.-3   | 0.699       | +0.2%  | 0.710       | +3.1%  | 0.718       | +3.6%   | 0.721       | +3.2%  |
> | Desc.-3  | 0.695       | -0.3%  | 0.656       | -4.7%  | 0.622       | -10.2%  | 0.691       | -1.1%  |
> | IID-5    | 0.701       | -      | 0.690       | -      | 0.694       | -       | 0.707       | -      |
> | Asc.-5   | 0.699       | -0.3%  | 0.697       | +1.0%  | 0.716       | +3.1%   | 0.721       | +2.0%  |
> | Desc.-5  | 0.695       | -0.8%  | 0.663       | -3.8%  | 0.642       | -7.5%   | 0.696       | -1.5%  |
>
>
> *IID baseline description*
>
> We have improved our description of the IID baseline in Section 3.2 as “IID denotes the i.i.d baseline where training data was uniformly and randomly drawn from the combined distribution of multiple levels, e.g. IID-2 denotes data is drawn from $\\{D_{1}, D_{9}\\}$”. And we further explained how we control the training process to ensure fair comparison using the vanilla SGD instead of momentum-based optimizers. We have updated Section 3.2 with red font accordingly.

---

> > ### Author Response · Authors · 2025-11-26
> >
> > *Performance gain*
> >
> > Figure 7 in the original paper (Figure 13 in the appendix of the revision), we show that the ascending curriculum improved the IID baseline by as much as 31.4% (Asc.-2 vs IID-2 with Medium training budget on the right subfigure). Additionally, we also made this clearer by adding Figure 5 and Table 1 in the appendix, where we show that curriculum learning from easy to hard with our defined transitional problem-based difficulty can yield a 3.6% performance gain over the IID baseline and 14.1% over the reversed curriculum in chess.
> >
> > In the math setting, we see that curriculum learning with transitional problems produces a significant increase in performance (over 5% in many cases) compared to learning from randomly sampled training problems.
> >
> >
> > *Training budget*
> >
> > Relevant results can be found in Figures 6 and 7 in the original paper (Figure 7 in Section 3.2 of the revision, as well as Figures 12, 13, and 17 in the appendix of the revision), where we show that the ascending curriculum outperforms baselines consistently across different training budgets.
> >
> >
> > *Length of the curriculum steps*
> >
> > Relevant results can be found in Figures 6 and 7 in the original paper (Figure 7 in Section 3.2 of the revision, as well as Figures 12, 13, and 17 in the appendix of the revision), and our newly added Table 1 in the appendix (the above table in this response). We did not find notable patterns in the effect of curriculum steps in chess. However, chaining up with 5 levels consistently outperforms 2 levels in math under various training budgets, suggesting that more curriculum steps may better help model learning in math. We would like to clarify that our goal to include multiple settings in terms of steps is to show the **consistency** of the advantage of ascending curricula over baselines, instead of finding the best hyperparameter setting. We have added the above discussion to Section 3.2 in the revision with red font.
> >
> >
> > *Distributional differences between puzzles and positions*
> >
> > Figures 10 and 11 in the Appendix complement Figure 3 in the main paper to show that, with or without distributional differences between puzzles and positions, transitional problems one level up from the model's competence consistently result in the highest performance improvement. Additionally, we have added Figure 5 and Table 1 in the paper to show that chaining up the level-up training strategy with an ascending curriculum consistently outperforms baselines. In particular, as shown in Table 1 in the appendix, distributional shifts between puzzles and positions do not affect such results. Interestingly, we do find that both training and testing with game-positions give the most advantage to the ascending curriculum compared with other distribution pairs.

---

### Official Review · Reviewer_Qvq3 · 2025-11-03

**Soundness:** 2
**Presentation:** 2
**Contribution:** 3
**Rating:** 2
**Confidence:** 4

**Summary:**

This paper proposes a novel framework for curriculum learning by introducing the concept of transitional problems - problems that show a monotonic decrease in difficulty as a model’s competence increases. The authors define formal criteria for identifying such problems within a model series and demonstrate the approach in two domains: chess and mathematical reasoning. Empirically, they find that fine-tuning models on transitional problems from the next competence level yields the most efficient performance improvements (“levels up” the model). Furthermore, curricula constructed from ascending sets of transitional problems outperform i.i.d. and descending curricula. The paper argues that this model-centric notion of difficulty provides a more principled basis for learner-specific curricula compared to human-centric or indirect proxy scoring approaches.

**Strengths:**

1. The idea of “leveling up” through transitional problems is intuitively compelling and well-motivated. The introduction of transitional problems as a means to measure difficulty relative to model competence is novel. It reframes curriculum learning around model-centric difficulty rather than human intuition.

2. The diagrams (especially Figure 1 and Figure 2) clearly convey the central idea and experimental setup, which help intuitively understand how transitional problems are defined and applied in curricula.

3. The chess experiments make sense and the results show consistent, convincing trends and strong correspondence with the hypothesis.

**Weaknesses:**

1. While the paper’s model-centric notion of difficulty is novel, it remains empirically motivated. It would strengthen the contribution to show why this definition is theoretically advantageous over other model-based measures (e.g., gradient norm, loss change, or C-score). Currently, the argument for why this formulation is superior is mostly intuitive rather than analytical or empirical.

2. Constructing a model series satisfying Definition 1 (monotonic increasing strength) is non-trivial in general settings, and the definition of transitional problems as those showing "monotonic decrease in difficulty as the model gets more capable" remains somewhat vague without clear guidelines for measuring capability. While chess serves as a strong example with Elo-based strength, extending this to a general ML method requires broader empirical validation across diverse tasks, model sizes, architectures, and domains beyond games and math.

3. The strength function appears ad-hoc and inconsistent across domains: Elo ratings in chess (external benchmark) versus validation accuracy in math (intrinsic metric). This raises concerns about generalizability and soundness. For other tasks, how should one determine an appropriate strength function?

4. The authors state that fine-tuning Qwen3-0.6B-Base on GSM8k improves from 3% to 40%, but the Qwen3 technical report lists 59.59% accuracy for this base model. Could you clarify how the Qwen3-0.6B checkpoints were constructed and why the model performance differs from reported numbers in the official technical report?

5. While the math experiments largely corroborate the trends from chess, some results appear inconsistent with the core hypothesis. For instance, in Figure 6's subplot for Level C3 → C4, the peak performance occurs at a lower difficulty level (Q-2) rather than the hypothesized "level-up" position (green line at Q-4). This deviation, with a noticeable drop in the curve around the expected optimum, suggests potential limitations in the method's robustness for math reasoning. Do you have hypotheses for these deviations?

6. While the paper reads smoothly overall, several minor presentation issues (e.g., inconsistent use of “Fig.” vs. “Figure,” missing figure references, and extra spaces) suggest the paper could benefit from a careful proofreading pass.

**Questions:**

Same as the Weakness section. I would be happy to discuss these points further during the rebuttal phase.

---

> ### Author Response · Authors · 2025-11-26
>
> *Advantage of our methodology over gradient norm, loss change, and C-score*
>
> Unlike existing methods for measuring model competence, our method not only looks at the learning signal provided by an example at a given point in time (i.e. a model at a particular strength), but also ensures that the example is relevant to the future learning trajectory of the model, i.e., can be solved consistently by stronger models. This information is not explicitly elicited by existing methods. Instantaneous methods such as the gradient norm and loss change are inherently biased towards the hardest samples, in contravention with the Zone of Proximal Development idea in human learning (Section 4). Aggregate methods such as C-score are highly expensive, and rely not on the importance of an example as a model gets stronger, but on information derived from weaker models that differ only in their initialization prior to training.
>
> Moreover, our method requires one forward pass through each example in the training dataset to identify the transitional problems on a predefined model series. Identifying a model series similarly requires only S backward passes and C\*S forward passes through the dataset, corresponding to training the model for S steps. Typically, S >> C, i.e., checkpointing is infrequent. This fixed, one-time work produces a permanent difficulty label for every sample, with no additional cost during curriculum learning. In contrast, gradient-based curricula (loss change, gradient norm, etc.) rely on one backward pass per training sample for every training step to identify the next batch of examples, resulting in S\*S backward passes and S\*S forward passes through the example. The considerable increase in expensive backward passes shows that gradient-based curricula are significantly more expensive than a curriculum over transitional problems, which amortizes the cost of identifying these problems.
>
>
> *Definition of transitional problems depends on the assessment of model capability, strength*
>
> We thank the reviewer for noting this ambiguity. Our original phrasing "monotonic decrease in difficulty as the model gets more capable" was intended to emphasize that problem difficulty is defined relative to model competence. However, we agree that this formulation was imprecise. In particular, problem difficulty is not changing, but its solvability is changing. We have revised the definition to be more concrete: transitional problems are "problems that exhibit a sharp transition in solvability across increasing competence levels: solved by models at or above a given level, but not by those below. These problems mark competence boundaries and yield an empirically grounded easy-to-hard partitioning of the training data distribution." We have updated the paper accordingly (marked in red).
>
>
> *Extension to general ML*
>
> We agree with the comment “extending this to a general ML method requires broader empirical validation across diverse tasks, model sizes, architectures, and domains beyond games and math”. Our paper suggests a direction that is promising and potentially useful across a broad set of domains and model settings, which will require a set of future papers to fully address. We note that our method for collecting our model series in the math experiments is fully agnostic to the choice of model, method, and dataset, and is valid as long as the objective can be ordered.
>
>
> *Generalizability of strength functions*
>
> We would like to clarify that both accuracy and Elo ratings are not domain-specific. We regard the choice of strength functions as a hyperparameter. For example, in LMArena [1], Elo ratings are used for ranking LLM response quality, which could be a valid choice for constructing a model series with increasing competence.
>
>
> *More stable math results* and *Qwen3 base result before fine-tuning*
>
> Please refer to General Response 2.
>
>
> *Typos*
>
> Thanks for your careful reading. We have made them consistent in the revision.
>
> [1] https://lmarena.ai/

---

### Official Review · Reviewer_XD5c · 2025-11-03

**Soundness:** 2
**Presentation:** 4
**Contribution:** 3
**Rating:** 4
**Confidence:** 3

**Summary:**

This paper introduces "transitional problems", a novel method for defining a curriculum based on a model's own learning trajectory. Problems are partitioned into discrete difficulty levels based on which model checkpoint in a series is first able to solve them, and consistently able to solve them after that. The authors show that for a model at competence level $i$, training on problems from level $i+1$ is the most efficient way to improve performance on other held-out problems from that same level. An ascending curriculum built on this principle shows modest gains over baselines.

**Strengths:**

- Novel approach to define sample difficulty, motivated by concepts from developmental psychology
- The experiments designed to test the "leveling up" hypothesis for a single $i$ -> $i+1$ step are compelling.

**Weaknesses:**

- The paper proves efficiency for single-step transitions but fails to provide any evidence that chaining these steps leads to a globally optimal or more efficient training process for achieving the best possible final model.
- The evaluation framework is self-referential. Showing that training on "level $k$ problems" makes a model better at "level $k$ problems" is an expected result and not a convincing demonstration of improved general task competence.
- The method requires an existing pre-trained oracle model to define the curriculum, making it inapplicable to the standard use-case of training a model from scratch.

**Questions:**

- The paper's claim relies on the assumption that a series of locally optimal steps ($i$ -> $i+1$) results in a globally optimal training path. Can the authors provide theoretical or empirical evidence to support this idea, as it is not self-evident?
- How can you be sure that the observed "leveling up" is not just an artifact of in-distribution generalization on a narrow data slice? Could you show, for instance, that training a model $M_i$ on $D_{i+1}$ leads to a greater performance increase on a general, unfiltered test set than training on $D_i$ or $D_{i+2}$?

---

> ### Author Response · Authors · 2025-11-26
>
> *No evidence that chaining up single steps leads to efficient training*
>
> Relevant results can be found in Section 3.2, where we explicitly show the results of the mentioned setting of “chaining these steps”. Training on the immediate next level of transitional problems most efficiently `levels up' the model, which induces a natural curriculum with ascending difficulty in terms of the transitional point. We observe consistent wins for the ascending curriculum over the i.i.d. baseline, and consistent losses for the descending curriculum for both domains (see Figures 5 and 7(a)). We also show that a curriculum based on transitional problems outperforms curricula trained on other sensible easy-to-hard orderings, such as question length or skills in math, or problem ratings in chess (see Figures 5 and 7(b)).
>
>
> *Self-referential: wins on transitional problems at a level depend on training from the same distribution*
>
> Our design uses a shifted test data distribution from the training set to ensure it’s non-trivial. Relevant results can be found in Figure 3, where we intentionally train on transitional **puzzles** and test on transitional **game-positions** to avoid observing an “artifact of in-distribution generalization”. Their difference is described at the end of Section 2.2.
>
> We emphasize that training on one level and helping the next is a natural but not obvious result. In particular, we found that the "level-up" result (training on the next level's problems for maximum gain) is consistent with the proposed model-centric transitional problem definition, but does not hold when difficulty is measured by external proxies like ELO rating in chess.
>
> Additionally, we add Figure 5 and Table 1 in the appendix to the paper to make this clearer, and we provide the table below for visibility, where $D^{puz}/D^{pos}$ denotes training on transitional puzzles, testing on transitional positions; $D^{pos}/D^{pos}$ denotes training on transitional positions, testing on transitional positions; and $R^{puz}/R^{puz}$ denotes training and testing on puzzles leveled by ELO ratings as the difficulty measure. This table shows that chaining up the level-up training strategy with an ascending curriculum consistently outperforms baselines. In particular, distributional shifts between puzzles and positions do not affect these results. Interestingly, we do find that both training and testing with game-positions give the most advantage to the ascending curriculum compared with other distribution pairs. We also observe that ascending curriculum learning with training and testing data leveled by ELO ratings does not outperform baselines, confirming the significance of the proposed model-centric difficulty measure based on transitional problems. We have improved Section 3.2 accordingly (marked in red).
>
> |          | $R^{puz}/R^{puz}$ | Impr   | $D^{puz}/D^{puz}$ | Impr   | $D^{pos}/D^{pos}$ | Impr    | $D^{puz}/D^{pos}$ | Impr   |
> |----------|:-----------:|:------:|:-----------:|:------:|:-----------:|:-------:|:-----------:|:------:|
> | IID    | 0.698       | -      | 0.688       | -      | 0.693       | -       | 0.699       | -      |
> | Asc.   | 0.699       | +0.2%  | 0.710       | +3.1%  | 0.718       | +3.6%   | 0.721       | +3.2%  |
> | Desc.  | 0.695       | -0.3%  | 0.656       | -4.7%  | 0.622       | -10.2%  | 0.691       | -1.1%  |
>
> In the math experiments, we conduct all of our evaluations on the held-out test set of the datasets we study. Thus, our performance increases are focused on model generalization in every setting. We also show that an ascending-difficulty curriculum defined based on the identified transitional problems outperforms both iid training and curricula based on other criteria, on a held-out set of test problems (see General Response 2 for a summary of these results). Hence the wins by training on transitional problems are not restricted to that distribution of transitional problems.
>
>
> *Results depend on existence of an oracle model*
>
> Please refer to General Response 1.

---

### Author Response · Authors · 2025-11-26

We thank the reviewers for noting the novelty of our method of defining a curriculum for model training, and our innovative definition of problem difficulty, as well as the experimental validation in two different reasoning domains, chess and math.

**General Response 1: Chicken-and-Egg Problem**

We also thank the reviewers for clearly articulating a motivational issue in the draft we originally submitted (the “chicken and egg” problem). We have now organized our motivation into three categories, which we present in the Introduction, and mention here as well.

First, transitional problems require a series of models of increasing competence, which in many cases already exist. The paper shows two examples (skill-adaptive models like Maia and the series of checkpoints that come with standard training runs of virtually any model), but there are several others as well. For example, frontier models typically form such a series, either within a single generation (e.g., Haiku, Sonnet, Opus models from Anthropic) or between generations (e.g., GPT-2, GPT-3, GPT-4, GPT-5). Thus, rather than requiring a brand new, perfectly trained oracle for this method alone, our methodology can make use of pre-existing model sequences. In this setting, we have conducted a new experiment that more concretely establishes the value of transitional problems. We take the strongest model in the series and train it only on problems that it currently fails on. We compare three models: random, “mistake” problems where the strongest model fails but some earlier model in the sequence succeeds, and “all mistake” problems where all models in the sequence fail. Training on “all mistake” problems yields the largest improvement on held-out problems where the strongest model initially fails (0.43 vs. 0.36 for “random”). This experiment shows that transitional problems define a directional training signal that can be used to improve the strongest model in the series. Finding this kind of “highest-gain” subset of problems lets us identify the most sample-efficient data to train on.

Second, there are settings where training the series of models first in order to identify transitional problems can make sense. One such setting is meta-learning, where we can amortize the up-front cost of training the initial series of models over many downstream models/tasks that make use of the resulting curricula. For example, Maia models can be easily fine-tuned toward individual chess-playing styles. By first training a base Maia, which is then fine-tuned to produce hundreds or thousands of individual models, each of which could benefit from the transitional problems and/or curricula that the original run produced, the up-front cost could be worth it. Another such setting is reusing transitional problems to train smaller or cheaper models, model variants with different architectures or training objectives, etc, or models on different but related datasets. Sharing cost across many downstream applications, and benefitting from the increased sample-efficiency of the learned curricula, can again overcome the up-front cost. We show an example of such a transfer in Section B.1 of Appendix B, where transitional problems identified in GSM8K are used to define a curriculum for training a model on a different math dataset (Orca), which shows clear benefits in training efficiency.

Third, the transitional problems themselves could be inherently valuable. Our methodology means that whenever an “oracle” is trained in a domain, we can then characterize the right questions to ask for any given level in that domain. This could be useful for learners (potentially other models or humans) that want to level up themselves. An example of this is the finding that the transitional problems that we find most useful in defining a curriculum for grade-school math do not correspond to the standard skill ordering of addition, subtraction, multiplication and division but instead depend on a particular combination of increasing question and answer length (see Section 3.3 and Figures 6, 8, and 9).

In summary, rather than being a “chicken-and-egg problem”, we view our methodology as enabling more sample-efficient and directed training for both the oracle model itself and for downstream/future models. The one-time cost of developing an oracle model may either be already paid (e.g. by existing models) or be amortized over benefiting many downstream learners.

---

> ### Author Response · Authors · 2025-11-26
>
> **General Response 2: New Results**
>
> 1. *Curriculum Learning Baselines*
>
> We conduct additional experiments to show that a curriculum based on transitional problems consistently outperforms curricula trained on other sensible easy-to-hard orderings, such as question length or skills in math, or problem ratings in chess (see Figures 5, 7(b), and Table 1 in the appendix). In particular, only when easy-to-hard curricula are defined with the transitional problem-based difficulty measure can they yield consistent wins over other strategies, such as random and reversed curricula, while aforementioned other difficulty definitions failed to show improvement with ascending curricula. Such results show that our transitional problem-based difficulty measure better captures the inherent model competence, which leads to successful leveled-up model performance with easy-to-hard ordered training data.
>
>
> 2. *Updated Math Results*
>
> Our math experiments rely on finding a model series starting from a base model that is not too good at grade-school level math, but can significantly improve when fine-tuned on the GSM8k training dataset. As identified by reviewer Qvd3, the base Qwen3-0.6B model is already very competent on the GSM8k dataset. The results presented in the report cited by the reviewer are for 4-shot CoT evaluation, and we find the following results with 0-shot CoT evaluation.
>
> Qwen3-0.6B-Base	: 	48.83%
>
> Qwen2.5-1.5B		:	28.29%
>
> Qwen2.5-0.5B		:	5.42%
>
> We hypothesize that the sharp drop in performance and subsequent increase when we train Qwen3-0.6B-Base is due to overfitting and then re-generalization. This caused our math experiments to have a high variance, and was not representative of the normal training of models. Thus, we re-run our experiments with Qwen2.5-0.5B to fairly evaluate a scenario where the model is indeed learning to perform better 0-shot, and replicate some experiments with Qwen2.5-1.5B to ensure that we do not over-rely on one model. Our revised experiments show many of the same trends as our initial experiments with Qwen3-0.6B-base, but with less variance between results and more consistent wins for curriculum learning over transitional problems compared to baselines.  Additionally, we add experiments comparing curriculum learning on transitional problems with curricula on randomly sampled problems from training dataset, and find that training on transitional problems leads to a significantly higher learning efficiency.
>
> We also evaluate a transfer learning setting obtaining neo-transitional problems on the Orca grade school math dataset via embedding similarity with GSM8k transitional problems. We show that training on these neo-transitional problems leads to similar improvements vs. baselines on the Orca dataset, resulting in a proof-of-concept to address the chicken-and-egg problem highlighted by multiple reviewers.
>
>
> 3. *Analysis of transitional problems*
>
> We have also added some analyses of the transitional problems in math. We measure the hardness of problems along five criteria: the length of the problem, the length of the solution, the number of reasoning steps (annotated by ‘<< >>’) in the gsm8k solution, the number of actual add/sub/mul/div operations in the problem, and the number of operations of a given skill (add/sub/mul/div) required to solve a problem. Our results (visible in Section 3.3 and Appendix B) show that transitional problems vary analogously to human notions of difficulty along three key measures- problem length, solution length, and the number of operations. Stronger models are also able to solve problems with more GSM8k-tagged reasoning steps on average. However, the composition of skills that the problem requires does not change on average, contravening the human notion of skill difficulty (e.g., division being more complex than addition). Thus, transitional problems offer a human-interpretable insight into model training but cannot be fully replicated simply by defining a predetermined ordering on these criteria.
>
> Notably, simply examining the set of all problems solved by a model does not produce the same human-interpretable notion of difficulty (appendix). This indicates that the consistency constraint on transitional problems (that a problem at level i can be solved by no model weaker than level i and every model at level i or stronger) is key to a robust and interpretable ordering of training problem difficulty.

---

### Author Response · Authors · 2025-12-03
**Final response to reviewers, and summary of rebuttals.**

## Main Contributions
- We introduce transitional problems, a novel model-centric definition of problem difficulty based on competence boundaries, which partitions training data into discrete difficulty levels.
- We demonstrate the level-up phenomenon: training on problems from the immediate next level yields the most efficient performance improvement, validated in both chess and math reasoning domains.
- We show that ascending curricula built on transitional problems consistently outperform IID training, reversed curricula, while easy-to hard curricula learning based on external difficulty proxies (question length, skills, ELO rating) do not work.
- We provide interpretable analyses showing that transitional problems align with human intuitions of difficulty along key dimensions while revealing that skill composition does not determine model-centric difficulty levels.

## Details

### Addressing the "Chicken-and-Egg" Problem (Reviewers XD5c, Qvq3, Ycia)
The most common concern was the apparent circularity of requiring a trained model series to define transitional problems. We have reorganized our motivation in the Introduction to address this concern with three complementary arguments:

- **Pre-existing model series.** Many domains already have sequences of models with increasing competence: skill-adaptive models like Maia, training checkpoints from standard runs, or frontier model families (e.g., Haiku/Sonnet/Opus, GPT-2/3/4/5). We conducted a new experiment showing that training the strongest model in such a series on partial transitional problems (where it fails and weaker models fail as well) yields larger improvements than training on random problems and "mistake" problems (where it fails but weaker models succeed), demonstrating that transitional problems provide a directional training signal for improving even the strongest model.


- **Amortization across downstream applications.** The up-front cost of identifying transitional problems can be amortized over many downstream learners: fine-tuned model variants, models with different architectures, or models trained on related datasets. We demonstrate this in Section B.1, where transitional problems identified on GSM8K transfer to define an effective curriculum on the Orca math dataset.

- **Intrinsic value of transitional problems.** The problems themselves characterize the right problems to be trained on at each competence level, potentially valuable for other learners (models or humans) seeking to level up. An example of this is the finding that the transitional problems that we find most useful in defining a curriculum for grade-school math do not correspond to the standard skill ordering of addition, subtraction, multiplication and division but instead depend on a particular combination of increasing question and answer length (see Section 3.3 and Figures 6, 8, and 9).

### Strengthening the Math Experiments (Reviewers Qvq3, zeBW)

Reviewer Qvq3 correctly noted that Qwen3-0.6B-Base already performs well on GSM8K (under 4-shot CoT evaluation). We clarified that our experiments use 0-shot CoT, where this model shows only 48.83% accuracy. However, to ensure our results are not artifacts of overfitting-then-regeneralization dynamics, we re-ran experiments with Qwen2.5-0.5B (5.42% base accuracy) and replicated key results with Qwen2.5-1.5B. Our revised experiments show consistent trends with reduced variance and clearer wins for curriculum learning on transitional problems. We also added a transfer learning experiment where neo-transitional problems on Orca, identified via embedding similarity with GSM8K transitional problems, yield similar improvements over baselines—providing a proof-of-concept for addressing the chicken-and-egg concern.

### Other Curricula as Baselines (Reviewers XD5c, Qvq3)

We conducted additional experiments comparing curricula based on transitional problems against curricula defined by other sensible difficulty measures (question length, skill type in math, ELO rating in chess). Only the transitional problem-based curriculum consistently outperforms IID and reversed curricula (Figures 5, 7(b), and Table 1 in the appendix). This confirms that our model-centric difficulty measure captures something fundamental about competence boundaries that external proxies do not.

### Out-of-Domain Generalization (Reviewer XD5c)

Reviewer XD5c raised concerns about in-distribution generalization. We emphasize that our chess experiments intentionally train on transitional puzzles and test on transitional game-positions (different distributions, described in Section 2.2), and the level-up phenomenon persists. In math, all evaluations use held-out test sets. Additionally, the level-up result holds specifically for our transitional problem definition but not when difficulty is measured by ELO ratings (Table 1), confirming that our model-centric measure is essential.

---

> ### Author Response · Authors · 2025-12-03
> **Final Response, continued**
>
> ### Computational Efficiency (Reviewers Qvq3, Ycia)
>
> We clarified that identifying transitional problems requires only one forward pass per example over a pre-defined model series (C checkpoints over S training steps, where S >> C). This one-time cost produces permanent difficulty labels with no additional overhead during curriculum training. In contrast, gradient-based methods (loss change, gradient norm) require one backward pass per example per training step, resulting in a considerable reduction in expensive backward computations (O(S²) backward passes versus our O(S) forward passes).
>
> ### Analysis of Transitional Problems (Reviewer Qvq3)
>
> We added analyses showing that transitional problems vary interpretably along problem length, solution length, and number of operations, which aligns with human intuitions of difficulty. However, skill composition (add/sub/mul/div) does not change across levels, indicating that transitional problems capture a notion of difficulty that cannot be replicated by simple predetermined orderings. Notably, examining all problems solved by a model does not produce this interpretable ordering; the consistency constraint in our definition is key.
>
>
> ## Summary of Revisions
>
> - Reorganized the introduction section to address the chicken-and-egg concern
> - New experiment showing transitional problems improve the strongest model in a series
> - Added curriculum learning baselines (question length, skills, ELO rating)
> - Re-ran math experiments with Qwen2.5-0.5B and Qwen2.5-1.5B for more stable results
> - Added transfer learning experiment (GSM8K → Orca)
> - Added analysis of transitional problem characteristics in Section 3.3 and Appendix B
> - Clarified definitions, baseline descriptions, and computational efficiency arguments
> - Fixed presentation issues (consistent formatting, typos)
>
> We believe these revisions substantially strengthen the paper and address the reviewers' main concerns. We thank the reviewers again for their engagement.

---

### Author Response · Authors · 2025-12-03
**Executive Summary of Final Remarks**

### Chicken-and-Egg Problem (Reviewers XD5c, Qvq3, Ycia)

We reorganized our motivation to show that (1) pre-existing model series are widely available, (2) the up-front cost can be amortized across downstream applications, and (3) transitional problems have intrinsic value for characterizing competence boundaries. We also conducted a new experiment demonstrating that transitional problems provide a directional training signal for improving even the strongest model in a series.

### Math Experiment Stability (Reviewers Qvq3, zeBW)

We re-ran experiments with Qwen2.5-0.5B and Qwen2.5-1.5B, showing consistent trends with reduced variance. We also added a transfer learning experiment (GSM8K → Orca) as a proof-of-concept for addressing the chicken-and-egg concern.

### Curriculum Learning Baselines (Reviewers XD5c, Qvq3)

We compared against curricula defined by question length, skill type, and ELO rating. Only transitional problem-based curricula consistently outperform IID and reversed curricula.
### Out-of-Domain Generalization (Reviewer XD5c)

Our chess experiments train on puzzles and test on game-positions (different distributions), and the level-up phenomenon persists.

### Computational Efficiency (Reviewers Qvq3, Ycia)

Identifying transitional problems requires O(n) forward passes (one-time cost), compared to O(n²) backward passes for gradient-based methods.

### Transitional Problems Characteristics (Reviewer Qvq3)

Transitional problems vary interpretably along problem length, solution length, and number of operations, but not skill composition, indicating they capture difficulty that cannot be replicated by predetermined orderings.

---

### Meta-Review · Area_Chair_icE3 · 2026-01-07

**Summary:**

The paper proposes a novel, model-centric notion of difficulty through “transitional problems” and demonstrates its use for curriculum learning in chess and mathematical reasoning.

**Reviewer Concerns:**

The reviewers generally acknowledge the conceptual novelty and clarity of the framework, but raise substantial concerns about the self-referential nature of the evaluation, the lack of theoretical grounding, limited empirical scope, and unclear generalizability beyond the two studied domains. Several reviewers point out that the method relies on the availability of a stronger or pre-trained model to define transitional problems, which limits its practical applicability and creates a chicken-and-egg concern that is not fully resolved, despite the additional experiments and clarifications in the rebuttal. . They also note that the empirical gains are sometimes modest or unstable (especially in the math setting), and that the current evidence does not convincingly show that chaining local “level-up” steps leads to globally better or more efficient training in general settings. While the authors have made a strong effort to address specific comments, including adding experiments and clarifying definitions, some core concerns about generality, soundness across domains, and dependence on specific model series remain open.

**Reviewer Scores:**

Therefore, based on the balance of strengths and unresolved weaknesses identified in the reviews, the overall recommendation is reject.

---

### Decision · Program_Chairs · 2026-01-26

Reject